

# Analyzing the Impact of Aeroelastic Model Fidelity on Control-Co Design Optimization of Floating Offshore Wind Turbines

Robert Behrens de Luna[1], Francesco Papi[2], David Marten[1], and Christian Oliver Paschereit[1]

[1]Chair of Fluid Dynamics, Hermann Föttinger Institute, Technische Universität Berlin, Müller-Breslau-Straße 8, 10623 Berlin, Germany.

[2]Department of Industrial Engineering, University of Florence, via di Santa Marta 3, 50139 Firenze, Italy.

**Correspondence:** Robert Behrens de Luna (r.behrensdeluna@tu-berlin.de)

**Abstract.** This work investigates the influence of aeroelastic modeling fidelity on design optimization of floating offshore wind turbines. To this end, the QBlade simulation environment was coupled to the Wind Energy with Integrated Servo-control wind turbine design and optimization framework. QBlade offers aerodynamic and structural models with varying levels of aeroelastic fidelity within a computationally efficient implementation. This enables time-domain optimization studies with levels of
aeroelastic fidelity that are currently often deemed unfeasible for such purposes due to the computational expense involved. Five fidelity combinations are considered, ranging from blade element momentum aerodynamics with torsion-constrained Euler–Bernoulli beams to lifting-line free vortex wake aerodynamics with fully populated Timoshenko beams. To assess how aerodynamic and structural modeling fidelity influences optimization outcomes, the parameters of the floating wind turbine controller are co-designed together with the floating substructure, a system typically considered less sensitive to aeroelastic
fidelity. The results show that controller tuning, structural load predictions and final design outcomes are all affected by the chosen fidelity level. Higher fidelity models broaden the design space through less conservative load estimates and variation in rotor operation, which in turn lead to more efficient platform designs. Increasing aeroelastic fidelity therefore improved the quality of the optimization results, albeit at the expense of higher computational cost.

## 1 Introduction

Floating offshore wind turbines (FOWTs) enable renewable energy generation in deeper waters where the installation of fixed-bottom systems is not feasible. Even though these systems have received significant attention in recent years, their complexity and cost continue to pose significant barriers to their widespread deployment. A comparison of capital expenditures (CapEx) between FOWTs and their fixed-bottom counterparts reveals that the floating substructure represents a major cost driver (Ghigo et al., 2020). This economic challenge must be addressed to enable broad deployment of floating wind and requires optimization
across all system components. Traditional sequential optimization strategies, in which each subsystem is optimized in isolation while others remain fixed, build in conservative assumptions at each design step, this can result in less efficient designs for highly coupled systems like FOWTs (Garcia-Sanz, 2019). In response, multidisciplinary design analysis and optimization (MDAO) has gained traction within the research community. A recent review by Ojo et al. (2022) summarizes the current state of MDAO in the context of FOWTs, outlines the key frameworks, trends as well as the remaining challenges in this field. MDAO



enables the simultaneous variation of design variables across interacting subsystems and thus accounts for coupled physical phenomena and potentially new, cost-effective design solutions (Martins and Ning, 2022). A sub-branch of MDAO is the idea of control co-design (CCD), where both the physical system and its controller are optimized and tuned simultaneously. CCD has shown significant potential to improve performance, reduce structural loads and lower total system cost by leveraging dynamic interactions between system and controller for complex systems (Garcia-Sanz, 2019). However, the benefits of CCD rely on the

accuracy of the underlying models, particularly for interactions between aeroelastic and control interactions. As turbine sizes increase, the structural flexibility becomes more pronounced. Hence, increasing the aeroelastic fidelity to accurately capture the complex interactions between aerodynamics and structural dynamics could proof essential (Veers et al., 2022).

   Current industry-standard simulation tools rely on low- to mid-fidelity models, such as the blade element momentum (BEM) method for aerodynamics and simplified beam theories to represent the blade structure. The latter do not fully account for cou-

pled interaction between the degrees of freedom of a beam element for structural dynamics. While computational efficiency is a key enabler of the BEM method, it requires numerous empirical corrections and can introduce significant uncertainty (Perez-Becker et al., 2020; Boorsma et al., 2020), especially in some of the dynamic conditions that large floating offshore turbines may encounter (Ramos-García et al., 2022; Schulz et al., 2025). As demonstrated in recent studies, employing lifting-line free vortex wake (LLFVW) methodologies (Behrens de Luna et al., 2024; Papi et al., 2024; Schulz et al., 2025) or advanced beam

models that resolve coupled structural dynamics (Papi et al., 2025) offers a pathway to more precise aeroelastic modeling, albeit at an elevated computational expense.

   Recent CCD studies for FOWTs used the Wind Energy with Integrated Servo-control (WEIS) framework and showed that simultaneous optimization of physical and control parameters can reduce structural mass, cost, or loads. For example, Zalkind et al. (2022) optimized controller parameters for the IEA 15 MW turbine on multiple floating platforms. Zalkind and Bortolotti

(2024) found a 2% lighter platform mass configuration for the IEA 22 MW VolturnUS-S when using simultaneous instead of sequential CCD. Abbas et al. (2024) found a levelized cost of energy (LCOE) reduction up to 4% when co-optimizing platform and controller. In the above-mentioned studies, OpenFAST was employed as the simulation tool, which was set up to use the BEM method aerodynamics and the ElastoDyn (NREL, 2025b) module for structural dynamics, which omits the blade torsional degrees of freedom. Outside of WEIS, frameworks such as (Yu et al., 2024) and (Bayat et al., 2025) have applied

CCD to FOWTs, but typically use reduced-order aeroelastic models to limit computational cost. The influence of increased aeroelastic fidelity on design outcomes, particularly in the context of control co-design, remains largely unexplored.

   To address this gap, this work builds upon the WEIS framework. Recent developments under the FLOATFARM (FLOAT-FARM, 2025) project have enabled the integration of the QBlade simulation code into WEIS. QBlade offers BEM and lifting-line free vortex wake aerodynamic models, as well as a structural solver that allows to select either Euler-Bernoulli, Timo-

shenko or Timoshenko fully populated matrix (FPM) beams, enabling fidelity variation within a unified workflow. The use of QBlade has been instrumental in maintaining the overall computational cost reasonable. The Timoshenko-FPM approach comes with very little overhead compared to simpler approaches. Moreover, while the lifting-line solver adds significant computational cost, QBlade allows for it's impact to be significantly reduced by exploiting Graphics Processing Units (GPUs). The contribution of this work to the literature is two-fold. First, the development and introduction of an optimization framework,





called QBtoWEIS, that builds on the capabilities of WEIS but allows direct comparison of model fidelity levels included in QBlade within CCD optimization problems. The second is the application and demonstration of this new tool to a modern and highly flexible reference turbine, such as the IEA 22 MW. This turbine was explicitly designed with passive load mitigation via bend–twist coupling in mind (Zahle et al., 2024a) and thus allows for a realistic assessment of aeroelastic fidelity effects on a on a complex system. In doing so, new insights into the trade-offs between aeroelastic model fidelity, computational cost and final optimum are provided. This work specifically addresses a gap that was recently outlined by a pool of experts in the wind energy community (Veers et al., 2022), which is the integration of advanced aeroelastic simulation tools into the design process and usage of increased fidelity models within optimization workflows.

In Section 2 we provide an overview of QBtoWEIS along with some theoretical background about the methods relevant for the optimizations carried out in this work. Section 3 is focused on the definition of the optimization problem and modeling considerations. In Section 4, the results are present and discussed.

## 2 Optimization Framework and Methods

### 2.1 Control Co-Design

Control Co-Design has recently drawn increased attention in the context of floating offshore wind turbines. The reason is that FOWT are highly coupled, nonlinear systems that consist of several sub-systems. The main ones being the wind turbine itself, the floating sub-structure, the mooring system and the servo control system. These sub-systems comprise multiple sub-systems in their own right. The wind turbine for instance is composed by the rotor-nacelle-assembly (RNA), the generator and the tower. Modifying only one of these components can have significant impact on the overall system behavior of the floating offshore wind turbine. If for instance the tower is to be optimized with the aim to reduce it's weight (and thereby cost), it is crucial to consider the rotor (1P) and blade-passing (3P) frequencies, to ensure that resonant frequencies to not coincide with operational conditions. Yet, not all interactions are as obvious as the tower-blade interaction and in order to avoid negative interactions between sub-systems, it is important to take a multidisciplinary design approach. CCD is a sub-category of MDAO, where the design or tuning of a controller takes place at the same time at which the physical system is designed or optimized. It represents a contrast to the sequential design approach, in which usually the controller is the final part of the design iteration. As is pointed out by Garcia-Sanz (2019), the CCD approach not only enhances the design process but also leads to improved system dynamics and controllability, ultimately resulting in lower costs and increased reliability.

### 2.2 QBlade in the WEIS Framework

WEIS (NREL, 2025f), short for Wind Energy with Integrated Servo-control, is a design framework developed by the National Renewable Energy Laboratory (NREL) for co-design of floating offshore wind turbines and their control systems. The framework integrates existing tools such as WISDEM (NREL, 2025g), OpenFAST (NREL, 2025c), ROSCO (NREL, 2025d) and pyHAMS (NREL, 2025i) into a unified workflow. Any wind turbine definition in the windIO (Bortolotti et al., 2025) format



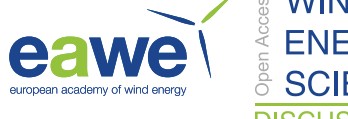

can be used as a starting point for an analysis or optimization. The framework is built on the OpenMDAO python library (Gray et al., 2019) and most of the tools are integrated as `explicit components`. This implementation enables efficient connection of corresponding inputs and outputs between tools in a python class that is often referred to as the `glue code`. The widely distributed QBlade simulation tool (QBlade, 2025) was expanded by a hydrodynamic module, which makes the

tool suitable to model floating offshore wind turbines. It has been validated and benchmarked against numerous other tools in varying conditions (Behrens de Luna et al., 2022; Behrens de Luna et al., 2024; Papi et al., 2024; Collier et al., 2024). Recent advancements in the FLOATFARM project (FLOATFARM, 2025) have now led to the integration of the multi-fidelity code QBlade into the WEIS framework. This coupling is henceforth referred to as QBtoWEIS (Behrens de Luna, 2025). Similarly to OpenFAST, QBlade runs non-linear time-domain simulations. In contrast, WEIS includes a reduced order frequency-domain

approach called RAFT. The aim of integrating QBlade in the WEIS framework is to expand the set of tools available for aero-servo-hydro-elastic analysis and to offer design engineers an alternative to OpenFAST and RAFT. This alternative enables increased-fidelity methods within design and optimization studies through its highly efficient implementation of the lifting-line method and Timoshenko-FPM beam elements. As shown in Fig. 1, an OpenMDAO component was created and embedded in the `glue code` of WEIS to efficiently manage the exchange of inputs and outputs between QBlade and the other WEIS

and WISDEM components, such as structural properties of the tower, floating platform natural frequencies, or pitch control tuning parameters, etc. An overview of the capabilities of WEIS, a discussion of available optimizers and related case studies can be found in (Zalkind et al., 2022; Zalkind and Bortolotti, 2024; WEIS Documentation, 2025). A more detailed analysis of the modeling approaches that are implemented in QBlade and their potential impact on design is provided in the following sections.

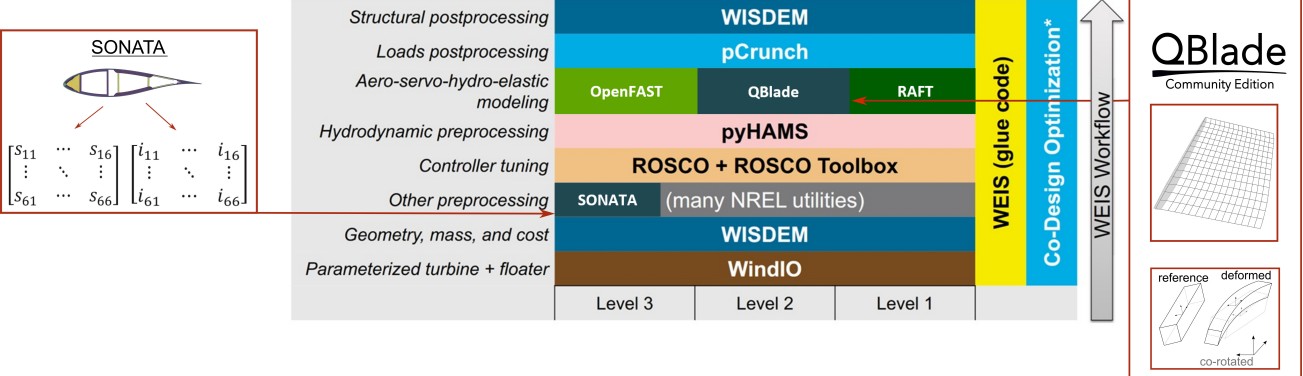

**Figure 1.** Tool stack of the QBtoWEIS framework. Newly integrated tools QBlade and SONATA (dark blue boxes) provide an alternative option for aero-servo-hydro-elastic modeling. QBtoWEIS extends the WEIS workflow (Zalkind et al., 2022). Figure adapted from (Zalkind and Bortolotti, 2024). Blade vortex lattice and beam sketch adapted from (Marten, 2020).

In order to obtain the equivalent beam parameters required for the Timoshenko-FPM beam model (i.e. the off-diagonal stiffness and inertia values), the Structural Optimization and Aeroelastic Analysis (SONATA) code was integrated into WEIS





as an additional OpenMDAO-component (Fig. 1). This step is necessary because PreComp, the current cross-sectional analysis tool available in WEIS, does not provide off-diagonal stiffness or inertia terms, nor the flapwise and edgewise shear stiffness (GA). SONATA is a cross-sectional analysis tool capable of deriving equivalent structural properties in the form of full $6 \times 6$

stiffness and inertia matrices for composite structures (Feil et al., 2020). SONATA was originally developed at the Institute for Rotorcraft and Vertical Flight, formerly Helicopter Technology Institute of the Technical University of Munich (RVF, 2025) and later adapted for wind turbine blade applications by NREL (NREL, 2025e). The tool builds on the open-source Python-based code for anisotropic beam analysis ANBA v4.0 (Morandini et al., 2010). The integration of SONATA into QBtoWEIS enables co-design of problems that include blade-related design variables, such as chord length, twist angle, or spar cap thickness,

with the structural blade modeled with Timoshenko-FPM beams. This capability of QBtoWEIS has already been used in a publication currently under review to investigate the influence of varying cross-sectional analysis capabilities on the blade design of a low-specific-power, 15 MW rotor (Papi et al., 2025).

## 2.3   Aerodynamic Wake Methods in QBlade

QBlade encompasses a traditional unsteady blade element momentum and a lifting-line free vortex wake method. According

to Perez-Becker et al. (2020), who systematically compared both wake methods in realistic conditions, the fatigue loads at various design relevant channels of an onshore turbine are overpredicted by the BEM method. Although the BEM method used for the comparison did not include dynamic inflow correction and a qualitative comparison with a BEM method that included this correction appeared to reduce the discrepancy between the models. In (Papi et al., 2024), the authors confirmed similar findings when the comparison was performed under floating offshore conditions. Based on these findings, a reasonable

assumption would be that including a lifting-line free vortex model in the design phase could allow for less conservative and more efficient design solutions. Boorsma et al. (2016) compared momentum-based and vortex-based methods, validating them with the New Mexico (Boorsma and Schepers, 2014) campaign. They observed better agreement between the higher fidelity method and the experiment in dynamic conditions, leading them to a similar conclusion concerning improved designs and reduced uncertainty if vortex methods are used. Schulz et al. (2025) systematically analyzed load amplitudes of the rotor thrust

force during fore-aft oscillation scenarios and considered frequency ranges that are typical for large FOWTS. They found that the dynamic wake effect, returning wake and unsteady airfoil effects result in discrepancies between the BEM and LLFVW methods in unsteady scenarios. All of the aforementioned phenomena are captured by the LLVW method, while the BEM method employs an empirical model to simulate dynamic wake and unsteady airfoil effects. However, the BEM method is incapable of capturing the returning wake event. Among these three unsteady effects, the dynamic wake effect is the most

prevalent, even in scenarios involving very slow fore-aft oscillation of the rotor. Even though the choice of aerodynamic method has implications for disciplines that do not focus on loads (e.g., wake propagation, wake breakdown, turbine-to-turbine interaction) as well, the focus of this work lies solely on loads. In this context the wake method largely differs in the way of how the wake-induced velocities are calculated.



### 2.3.1 Blade Element Momentum Theory

The blade element momentum theory has been the industry standard to simulate loads on a wind turbine for several decades. Its fundamental algorithm is detailed in widely used textbooks (Hansen, 2008; Burton et al., 2001). Essentially, as the name implies, the theory combines momentum and blade element theories to find two expression of the aerodynamic thrust and two expressions of the aerodynamic torque, which are iterated to solve for the axial induction factor $a$ and the tangential induction factor $a'$. By finding consistent values of $a$ and $a'$ from both theories, an equilibrium state is established. Over the years, many

engineering models have been applied to the BEM algorithm to address certain shortcomings that are related to assumptions that are made by the BEM method (Branlard et al., 2022; Madsen et al., 2020; Snel and Schepers, 1995; Buhl, 2005). The implementation in QBlade closely follows the unsteady polar BEM method as described by Madsen et al. (2020).

### 2.3.2 Lifting-Line Free Vortex Wake

The LLFVW wake method relies on the the basic lifting-line theory developed by Prandtl and Lanchester. The bound vorticity

of the blade is found by iteratively solving the Kutta-Joukowski theorem using estimates from 2D airfoil theory, typically derived from tabulated polar data.

$$L = \rho v_{tot} \times \Gamma \quad \text{and} \quad L = C_l(\alpha)\frac{1}{2}\rho * v_{tot}^2 c \tag{1}$$

L is the lift force per blade section, $\Gamma$ the corresponding circulation, $\rho$ the density, $v_{tot}$ the total velocity and $C_l$ the lift coefficient that depends on the angle of attack $\alpha$ . The total velocity is composed by

$$v_{tot} = v_\infty + v_{mot} + v_\Gamma \tag{2}$$

with $v_\infty$ being the free stream velocity, $v_{mot}$ the motion velocity of the blade and $v_\Gamma$ the induced velocity by the wake. The latter velocity component can be calculated via the Biot-Savart law that evaluates the contribution of all surrounding vortex lattices on to the evaluation point (van Garrel, 2003; Perez-Becker et al., 2020; Marten, 2020).

$$\boldsymbol{v}_\Gamma = -\frac{1}{4\pi}\int\frac{\Gamma\left(\boldsymbol{x}_p - \boldsymbol{x}\right)\times d\boldsymbol{l}}{|\boldsymbol{x}_p - \boldsymbol{x}|^3} \tag{3}$$

In Eq. (3), $\boldsymbol{x}_p$ is the location where the induced velocity is being evaluated, $\boldsymbol{x}$ the position vector of any given vortex element and $d\boldsymbol{l}$ the vectorized length of a vortex element. Solving the Biot–Savart law makes this method more computationally demanding than the BEM method because the full wake is taken into account when evaluating the induced velocity at a given position. Thus, to evaluate the induced velocity at a blade section, the contribution of each wake element must be calculated explicitly. Furthermore, to convect the wake, the local induced velocity at each element is calculated to determine the convection velocity.

This treatment allows the wake to develop freely. Problems like these are referred to as $\mathcal{O}(N^2)$ problems. In QBlade, an OpenCL-based GPU parallelization of the Biot–Savart evaluations enables a considerable decrease in computation time. Each vortex element is encoded into OpenCL vector primitives and distributed across a large number of GPU cores Marten (2020). As shown in benchmark tests in Fig. 2, the GPU implementation is about two orders of magnitude faster compared to a single-core CPU evaluation, making it viable for load simulations and design studies.



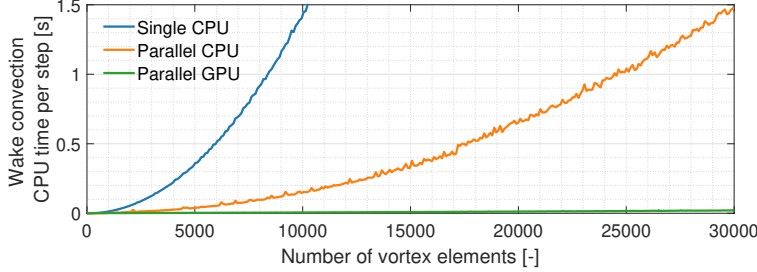

**Figure 2.** Scaling of LLFVW implementation in QBlade when GPU acceleration is used, from (Marten, 2020).

## 2.4 Structural Model in QBlade

With the ever increasing sizes of wind turbines, the method to capture the structural dynamics of a wind turbine and in particular the blades is critical for accurately predicting aeroelastic loads. In floating offshore wind in particular, the coupling between aerodynamic forces and the flexible structural response is amplified by platform motions and low-frequency excitations. The latest generation of open-source research wind turbines (the IEA Wind-22 Megawatt Offshore Reference Wind Turbine) underlines the importance of the structural model, since during its design, a link between the flapwise-bend and torsional degrees of freedom, often referred to as bend-twist or shear-twist coupling, was considered to passively reduce the loads in high thrust operation conditions (Zahle et al., 2024a). In order to accurately capture this effect, the structural model of a simulation tool should ideally resolve the coupled dynamics between bending, shear and torsion along the blade span (Papi et al., 2025).

The current state-of-the-art to model a structural wind turbine is to assemble it in a multi-body formulation in which the tower and the three blades (in some codes also the drive train) are are connected via joints and constraints, (Guo et al., 2024). While outlining the development of yet another multi body framework, Guo et al. (2024) gives a good overview of available structural solvers in wind turbine simulation tools and their assumptions. One distinction between all the FEA models is the beam model in the multi-body formulation. OpenFAST can either rely on the ElastoDyn(NREL, 2025b) or BeamDyn (NREL, 2025a) modules. The former evokes Euler-Bernoulli beams but assumes a set of prescribed degrees-of-freedom through a modal reduction approach (Branlard and Geisler, 2022) and, critically, neglects the torsional degree of freedom. BeamDyn on the other hand uses geometrically exact beam theory (Hodges, 2006). However, its computational speed makes it currently unfeasible for optimization or design tasks.

### 2.4.1 Multi-Body representation in QBlade

QBlade, like OpenFAST, models the turbine structure as a multi-body system. In order to do so, the open-source multi-physics engine Project::Chrono (Tasora et al., 2016) is integrated with the code. Each blade is discretized using a series of one-dimensional beam elements arranged in a co-rotational formulation, enabling the accurate capture of large displacements and nonlinear geometric effects (Marten, 2020). Unlike in ElastoDyn, the torsional degree of freedom is available. To resolve the influence of shear and anisotropic composite behavior, QBlade accommodates multiple beam models, namely Euler-Bernoulli,





Timoshenko or Timoshenko-FPM. The Timoshenko-FPM beam resolves all six degrees of freedom with full cross-couplings
in the mass and stiffness matrices. This enables the representation of complex structural behaviors, such as shear-twist and
shear-twist coupling. The computational overhead of using Timoshenko-FPM beams in QBlade is modest, making it suitable
for computationally intensive tasks such as optimization and design studies. A recent comparison by (Papi et al., 2025) em-
ployed the QBtoWEIS framework, leveraging the integration of SONATA, to assess the influence of different beam modeling
approaches on the aeroelastic response of long and flexible turbine blades and their influence on blade optimization. The study
highlights how variations in structural fidelity between Euler-Bernoulli and Timoshenko-FPM beam formulations can lead to
notable differences in load prediction, deformation and design outcome.

## 3  Optimization Problem, Modeling Considerations and Computation

### 3.1  Optimization Problem

To assess the impact of aeroelastic modeling fidelity on CCD and MDAO, an optimization problem for the IEA Wind-22
MW Offshore RWT mounted atop the semi-submersible floating structure that builds on the architecture of the VolturnUS-S
platform (Zahle et al., 2024a) is formulated. As shown in Fig. 3, the platform consists of a center column that connects to the
tower and three further outer columns, all connected via cylindrical members.

(a) (b) (c)

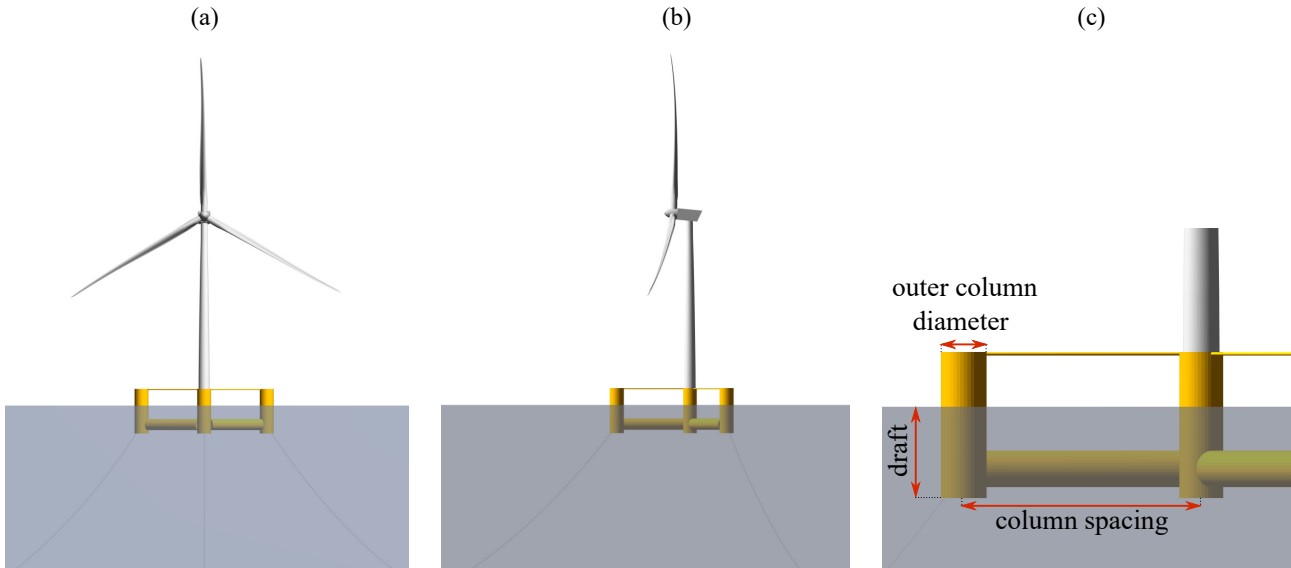

**Figure 3.** IEA 22 MW RWT baseline configuration, rendered in QBlade. (a) front view, (b) side view, (c) design variables of the floating
substructure.

The optimization problem is inspired by the problem defined by Zalkind and Bortolotti (2024) and aims to minimize damage
equivalent loads at the tower base by varying geometric parameters of the floating substructure and tuning parameters of the





**Table 1.** Main system properties of the IEA 22 MW RWT, see Zahle et al. (2024a) for a complete definition.

| Property | Unit | Value |
|---|---|---|
| Turbine Rating | MW | 22 |
| Wind Class | - | 1B |
| Rotor Diameter | m | 284 |
| Hub height | m | 170 |
| Blade prebend | m | 8.4 |
| Hub system mass | t | 120 |
| Tower mass | t | 1.574 |
| Blade mass | t | 82.301 |
| Rated Wind Speed | $\frac{m}{s}$ | 11 |
| Rated RPM | $\frac{1}{min}$ | 7.061 |
| Tip Speed Ratio | - | 9.5 |

turbine controller, under a set of constraints. This design problem was selected as a representative multidisciplinary case. Wind turbine rotor and substructure are often designed in isolation from one-another, frequently by different companies relying on separate tool chains. In practice, the exchange of information between these tools is limited to high-level data. Thereby, the impact of rotor aeroelastic modeling fidelity in substructure design is often disregarded. For this reason, this work aims to investigate how different levels of rotor aeroelastic fidelity influences the overall design optimization of floating offshore wind

turbines, with particular attention to the coupled rotor–substructure dynamics. Controller tuning is included in the design space since the baseline controller settings may not remain stable when the platform geometry changes. Inappropriate tuning could cause or amplify resonance effects and ultimately increase tower base DELs, even if the substructure geometry is improved. In addition, including the controller enables exploration of design configurations that might otherwise be infeasible or sub-optimal if the control system remained fixed. More specifically, as shown in Table 2, the optimizer can vary the platform draft,

outer column diameter and column spacing. The remaining design variables belong to the pitch control subsystem of the servo controller. These include the floating feedback gain, $k_{float}$ and the low-pass filter cut-off frequency $\omega_{float}$ which are primarily tuned to mitigate the negative aerodynamic damping problem (Skaare et al., 2007; Jonkman, 2010; Larsen and Hanson, 2007) through the parallel compensation logic (van der Veen et al., 2012; Abbas et al., 2022). To briefly summarize the logic, the tower-top acceleration is low-pass filtered and integrated to generate a noise-reduced estimate of the tower-top velocity. This

signal is then scaled by the gain $k_{float}$ and fed back to the pitch controller. Also part of the optimization are the closed-loop bandwidth of the pitch controller $\omega_{pc}$, which determines how quickly the controller responds to disturbances and the damping ratio $\zeta_{pc}$, which characterizes how oscillations in the closed-loop system decay. Both $\omega_{pc}$ and $\zeta_{pc}$ have three control points at wind speeds 12, 17 and 23 m/s, allowing the pitch controller's behavior to vary across the region III.





The multidisciplinary workflow used to evaluate each design is summarized in the extended design structure matrix (XDSM)
diagram of Fig. 4. Because varying the aeroelastic fidelity required only modifications to the QBlade turbine definition, the
workflow remains identical across cases except for changes to either the structural beam model (`'BEAMTYPE'`) or the aerody-
namic wake model (`'WAKETYPE'`) under the QBlade object within in the modeling options of the WEIS problem definition.
This setup allows a controlled comparison of the influence of aeroelastic fidelity on the optimization results, with all other
components of the loop unchanged. In this way, any differences in the optimal solutions can be attributed to the fidelity level
of the aeroelastic model.

**Table 2.** Design Variables and Constraints

| (a) Design Variables | | | | (b) Constraints | | |
|---|---|---|---|---|---|---|
| Design Variables | Lower Bound | Upper Bound | | Constraints | Lower Bound | Upper Bound |
| Draft | 35 m | 20 m | | Max. platform mass | - | initial mass |
| Outer column diameter | 10 m | 16 m | | Min. AEP | initial AEP | - |
| Column spacing | 60 m | 67.5 m | | Max. platform pitch | - | 6.5 m |
| PC natural frequency ($\omega_{pc}$) | 0.025 rad/s | 0.5 rad/s | | Max. nacelle acceleration | - | 2.85 m |
| PC damping ratio ($\zeta_{pc}$) | 0.5 rad/s | 2.5 rad/s | | Max. generator overspeed | - | 28.5% |
| Fl. feedback gain ($k_{float}$) | 8 s | 20 s | | Max. avg. pitch travel | - | 0.085 deg/s |
| Fl. feedback cut-off freq. ($\omega_{float}$) | 0.00001 rad/s | 0.5 rad/s | | floater heave period | 14 s | 18 s |
| | | | | floater pitch period | 20 s | 22 s |

As shown in Fig. 4, the constraint optimization by linear approximation (COBYLA) (Powell, 1994)) is used[1]. The optimizer
updates the design variables and starts an analysis chain. The platform geometry is assembled, component properties (e.g.,
blade structure, platform mass) are derived and steady-state Cp-Ct surfaces are generated in WISDEM and SONATA. The
controller is tuned in ROSCO and the aero-servo-hydro-elastic simulation in QBlade is run using either BEM or LLFVW
wake models and Euler–Bernoulli or Timoshenko-FPM beam representations . The resulting time series are post-processed in
pCrunch (NREL, 2025h) to compute tower-base DELs and evaluate constraints (e.g., platform pitch, annual energy production
(AEP), heave or pitch periods), which are returned to the optimizer to close the loop.

## 3.2 Modeling Considerations

The QBlade models used in all five optimizations were identical, only chosen wake and beam models differed between them.
The floating substructure model was derived from the windIO definition provided by Zahle et al. (2024b) and was assumed to be
rigid. Furthermore, the strip-theory approach described in Zahle et al. (2024a) was applied and the hydrodynamic coefficients
($C_a$, $C_d$ and $C_p$) were set accordingly. In order to verify the natural frequencies of the assembled FOWT and its dynamic
response to wave excitation, Figure 36 of (Zahle et al., 2024a) was used. The response amplitude operators (RAOs) were

---

[1]Specifically, the implementation in of (Johnson, 2007)





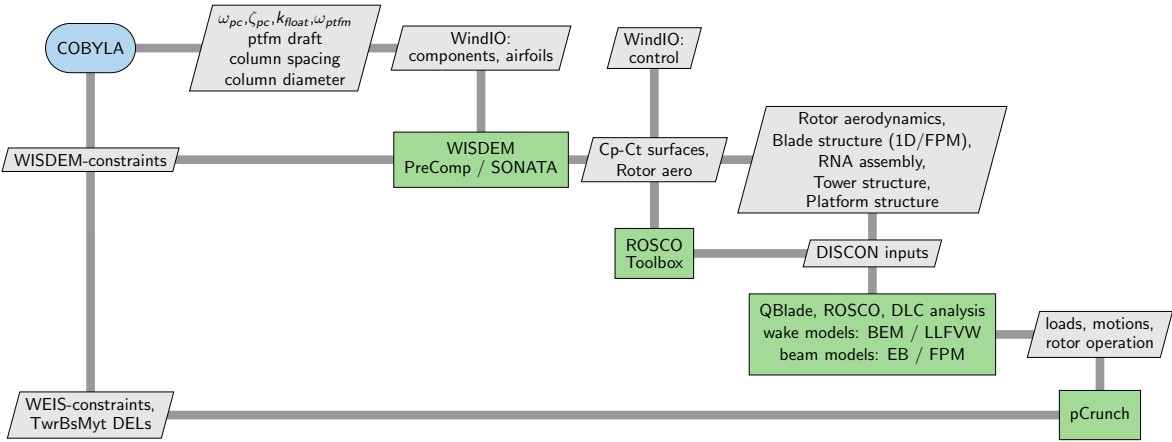

**Figure 4.** XDSM diagram of the core optimization problem.

derived from QBlade simulations using white-noise waves, following the procedure described by Ramachandran et al. (2013).
QBlade-specific parameters were defined in the `modeling_options.yaml` file. To reduce the influence of transients, the initial conditions were set to an 11 m surge displacement and a $1°$ platform pitch angle. Initial rotational speed and blade pitch are wind speed specific and provided by the WISDEM module RotorSE. The OYE dynamic stall model was activated with a time constant of $\tau = 8$. Regarding hydrodynamics, distributed buoyancy was enabled, the MacCamy–Fuchs correction was applied and Wheeler stretching was selected. The wave field was discretized into 500 linear wave components with
equal frequency spacing. Design load case (DLC) 1.1, which is based on the IEC standards (International Electrotechnical Commission, 2019), was used for a class IB turbine. The metocean conditions corresponding to an offshore location west of the Isle of Barra in Scotland (Papi et al., 2022), which are particularly rough offshore conditions, were used to define the sea state under normal conditions. The operating range of the wind turbine was covered with 10 wind speed bins[2] and for each wind speed 6 seeds were simulated, resulting in 60 simulations per iteration. To reduce the influence of transients, a 250 s transient
time was defined. The analysis time for each simulation was 600 s resulting in 850 s total simulated time per simulation. All results shown in this work used QBladeEE v2.0.9 and QBtoWEIS v1.1.0.

### 3.3 Computational Considerations and used Infrastructure

Since QBtoWEIS is parallelized, the number of simulations per iteration can be set without significantly impacting the overall runtime, as long as sufficient Central Processing Unit (CPU) cores and GPUs are available. This is particularly the case for
BEM simulations, which only evaluate on the CPUs. In contrast, LLFVW simulations rely on GPUs and oversubscribing a single device can increase computational time significantly. Hence, for this work, a limit of 60 simulations per iteration was applied to limit overall runtime. BEM simulations were executed on the *CPU CLX* partition at the national high performance

---

[2]5, 7, 9, 11, 13, 15, 17, 20, 23 and 25 m/s





computing center at the Zuse Institute Berlin (NHR@ZIB). Each compute node is equipped with two Intel Xeon Cascade Lake
Platinum 9242 processors, with 96 compute cores and 384 GB RAM. LLFVW simulations were run on the *GPU A100* partition
at NHR@ZIB, which consists of nodes with two Intel Xeon Ice Lake Platinum 8360Y processors (72 cores total), 1 TB RAM
and four NVIDIA A100 GPUs (80 GB HBM2 each). The physical time per iteration as well as charged core hours per node
can be found in Table 3.

**Table 3.** Required duration and cost per iteration on the NHR@ZIB HPC.

| Aero model | Struct model | Duration | Core hours |
|---|---|---|---|
| BEM | GJ (10x) | 20 min | 32 |
| BEM | Euler–Bernoulli | 20 min | 32 |
| BEM | Timoshenko–FPM | 24 min | 38 |
| LLFVW | Euler–Bernoulli | 40 min | 400 |
| LLFVW | Timoshenko–FPM | 40 min | 400 |

## 4   Results and Analysis

This section presents and analyzes the results of the main optimization problem, which is aimed at reducing the damage
equivalent loads at the tower base[3]. The problem is run with five different aeroelastic model fidelity combinations (see Table 4)
to assess their influence on optimization outcomes. The first combination, which constrains the torsional degree of freedom,
represents the fidelity level provided by OpenFAST combined with ElastoDyn — the current state of the art in WEIS. This
analysis is followed by a discussion in Subsection 4.3, which introduces a variation of the optimization problem with the
levelized cost of energy as a merit figure. This alternative formulation was motivated by initial findings, which suggested it
could provide additional value to this work.

**Table 4.** Levels of aeroelastic fidelity compared in this study, ordered from low to high fidelity (in relative terms).

| Aerodynamic model | Structural model | Abbreviation | Fidelity level |
|---|---|---|---|
| Blade Element Momentum | Euler–Bernoulli, GJ×10 | BEM GJ10 | Low |
| Blade Element Momentum | Euler–Bernoulli | BEM EB | Low–Medium |
| Blade Element Momentum | Timoshenko–FPM | BEM FPM | Medium |
| Lifting-Line Free Vortex Wake | Euler–Bernoulli | LLFVW EB | Medium–High |
| Lifting-Line Free Vortex Wake | Timoshenko–FPM | LLFVW FPM | High |

---

[3]The load channel is defined by default in WEIS as the Frobenius norm (based on NumPy's `numpy.linalg.norm()` function (Harris et al., 2020)) of
the tower base moments in fore-aft, side-side and torsion. The fore-aft moment is the primary contributor to the metric.



## 4.1 Baseline Comparison

Before the optimization results are analyzed, a statistical comparison of the results from the baseline (iteration 0) is provided in the following two subsections to highlight differences caused by the varying levels of aeroelastic model fidelity on the same FOWT.

### 4.1.1 Statistical Comparison

Figure 5 displays statistical metrics across different levels of aeroelastic fidelity for the baseline iteration. Each color represents a distinct fidelity level. The horizontal bars indicate mean values, vertical lines the standard deviation and the upward and downward triangles reflect the maximum and minimum values observed across all six seeds for a given wind speed bin. The low-speed shaft force in downwind direction (a), which serves as a measure of total rotor thrust, reveals clear discrepancies between the fidelity levels, which can be traced back to modeling fidelity. Several tendencies are consistent within regions II and III of the power curve:

 (i) Cases with torsion-constrained or full Euler-Bernoulli beam models (*EB and GJ10) consistently show higher mean thrust levels compared to those with a Timoshenko-FPM beam.

 (ii) Cases using the LLFVW wake model show reduced variation (i.e., smaller standard deviations and min-max ranges)

 (iii) Cases using the LLFVW wake model show higher thrust levels relative to BEM simulations using equivalent beam models.

 (iv) In region III, results across all fidelity levels align more closely.

The cause for (i) can be traced back to blade twist behavior, as shown in subfigure (b). The tip of the blade for the GJ10 model undergoes almost no twist. In contrast, the Euler-Bernoulli beam models show a steady increase in twist (towards feather) up to rated wind speed at a mean value of around 3.5 degrees. Since cross-coupling terms are not included in this model, the rotation results from aerodynamically induced torsion only. The FPM-based models show a larger tip rotation of up to 5.5 deg, before decreasing back to approximatly 3.5 degrees in region III. This increased rotation around rated wind speed can be attributed to shear-twist coupling, caused by the interaction between torsional and flapwise-shear degrees of freedom captured in the Timoshenko-FPM beam representation. As a consequence of varying spanwise twist between the levels of fidelity, the local angle of attack is larger in models with reduced twist. A higher angle of attack in turn, leads to larger aerodynamic forces in both the tangential and normal directions, which results in increased thrust and torque. Furthermore, the behavior of reduced variation and increased thrust in LLFVW cases can be explained with several differences in wake modeling. First, the LLFVW model captures dynamic events (e.g., gusts, floater motion) and their influence on the wake explicitly through the shed vorticity. The BEM models in contrast require the dynamic inflow corrections (Mancini et al., 2023). Second, LLFVW tracks induction locally along the blade. In contrast, classical BEM assumes uniform induction within each annulus. Even though the polar grid BEM extension allows for azimuthal variation, this treatment remains sectional and does not provide the fully local





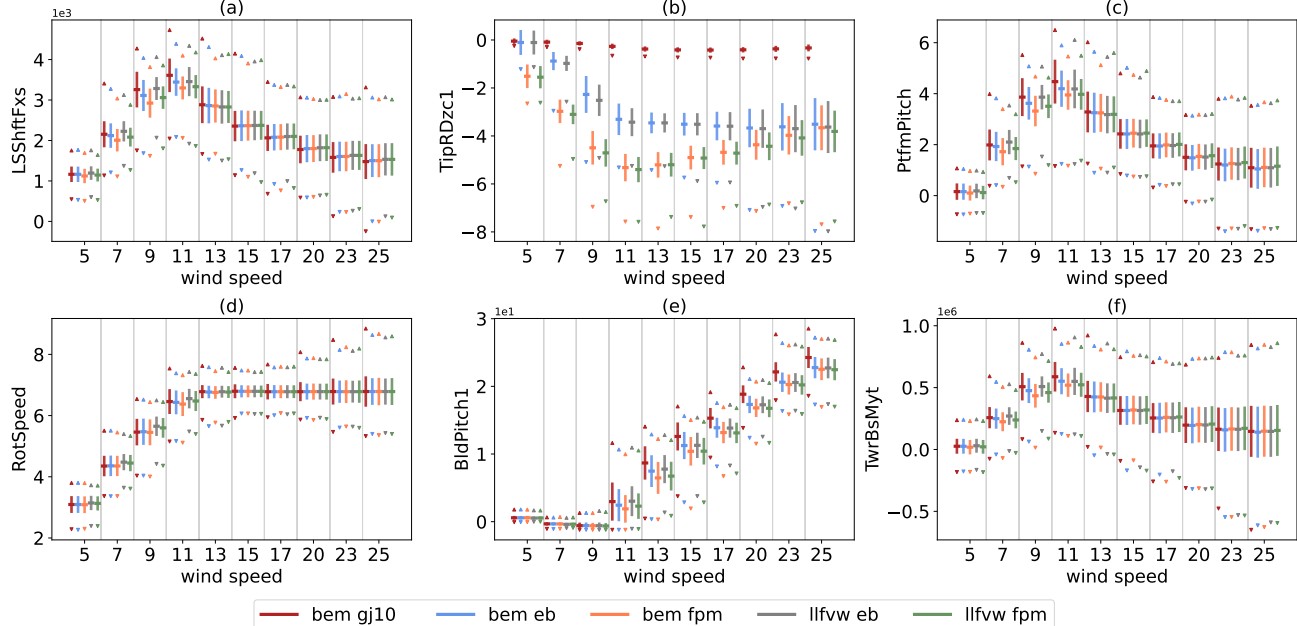

**Figure 5.** Comparison of statistical metrics for the baseline configuration (iteration 0) of each fidelity level. (a) low-speed shaft force in downwind direction [kN], (b) rotation of blade tip [deg], (c) pitch angle of the platform [deg], (d) rotational speed [1/min], (e) blade pitch angle [deg], (f) tower base fore-aft moment [kNm]. For the sake of readability, the maximum values for subplot (b) are not shown.

resolution of induction that is obtained with LLFVW. As a result, the inflow is less accurately represented, typically yielding more fluctuating angle-of-attack values, as discussed by Boorsma et al. (2016). Concerning (iii), the LLFVW methods tends to predict lower overall rotor induction compared to BEM, leading to higher axial wind velocities in the rotor plane. As a result,

the controller's wind speed estimator drives the system toward a higher rotational speed in order to maintain an optimal angle of attack. This in turn leads to increased aerodynamic loading and hence larger thrust forces. Finally, the reduced difference between models in region III is due to pitch controller activation above rated windspeed. Here, the controller prevents the rotational speed to overshoot the rated speed and reduces the aerodynamic force by pitching towards feather. As shown in subfigure e, the torsion-constrained BEM GJ10 model requires the highest blade pitch angles, followed by the Euler-Bernoulli

beam models and finally the ones with Timoshenko-FPM beams. For a an equivalent beam model, the LLFVW cases require more pitch actuation than their BEM counterparts, again reflecting the higher aerodynamic loading described earlier. However, pitch standard deviation is lower in LLFVW simulations. Platform pitch and tower base fore-aft moment (subfigures c and f) are strongly correlated with rotor thrust and, above region III, blade pitch.





#### 4.1.2 Time Domain Analysis

Figure 6 presents a selection of channels from a full run of a single seed at an 11 m/s average wind speed. This operating point is well suited to compare the different cases, as the wind speed frequently crosses the rated threshold, triggering transitions between torque and pitch control. Within the shown time window, the wind velocity dips noticeably below rated on three occasions (around 400 s, 550 s and 700 s), each time causing the generator torque (subplot e) to drop from its rated value. Two trends emerge:

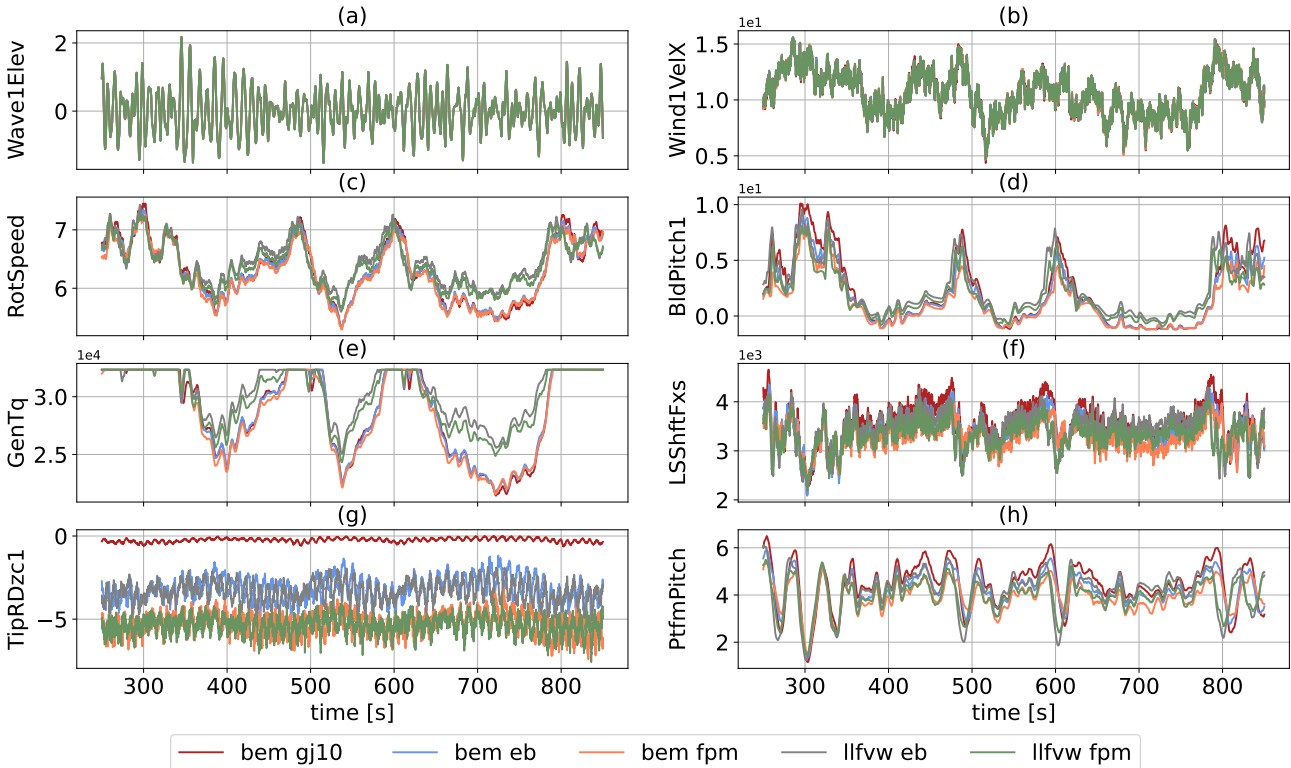

**Figure 6.** Comparison of baseline timeseries for a case with 11 m/s of each fidelity level. (a) wave elevation at reference point [m], (b) wind velocity at hub height [m/s], (c) rotational speed [1/min], (d) blade pitch angle [deg], (e) generator torque [kNm], (f) low-speed shaft force in downwind direction [kN], (g) rotation of blade tip [deg], (h) pitch angle of platform [deg].

(i) simulations using the BEM wake model exhibit a stronger torque reduction compared to the LLFVW cases and require more time to recover to rated torque;

(ii) with the aerodynamic wake method equal, the FPM-based structural models generally produce lower torque than their EB counterparts.





The first trend is consistent with the differences in axial induction predicted by the two wake methods, as discussed in
Section 4.1.1. The second is due to blade twist behavior (subplot g), where the respective beam model affects the torsion at
the blade tip. The blade pitch time series (subplot d) shows actuation whenever wind speed exceeds rated. As expected, the
LLFVW cases initiate pitching slightly earlier, in line with their earlier recovery of generator torque to rated levels. Further,
a small offset is visible between the LLFVW and BEM cases, which persists even during periods where blade pitch nears its
saturation angle (e.g., around 700 s). Finally, the thrust and platform pitch responses (subplots f and h) reflect the aeroelastic
fidelity effects identified previously:

(i) LLFVW models predict higher thrust than BEM;

(ii) EB models predict higher thrust than FPM;

(iii) the torsion-constrained GJ10 case yields the highest thrust among all.

## 4.2 Convergence Trends

This section presents the convergence trends of the optimization process, focusing on the merit figure (damage equivalent loads
at the tower base), the constraints and the design variables.

### 4.2.1 Merit Figure

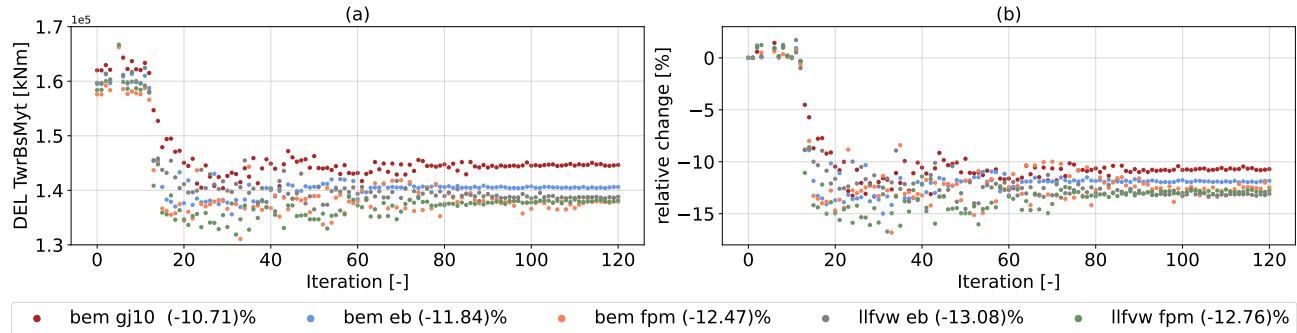

**Figure 7.** Convergence of the tower base bending moment DEL. (a) shows the absolute DEL values and (b) the relative change of each fidelity
level with respect to its initial iteration. The maximum iteration limit was set to 120, where all optimizations appear to have converged.

All fidelity levels achieve a substantial reduction in DEL at the tower base over the optimization process, though the final
achieved optima vary. The BEM GJ10 model converges to the least favorable solution in terms of percentage point reduction,
while the BEM EB model settles slightly lower. The remaining three configurations (BEM FPM, LLFVW EB and LLFVW FPM)
achieve the largest relative reductions of approximately 12.5–13.1%. This trend suggests that increasing aeroelastic fidelity
improves not only the accuracy of load prediction but also enables more effective optimization.



### 4.2.2 Design Variables

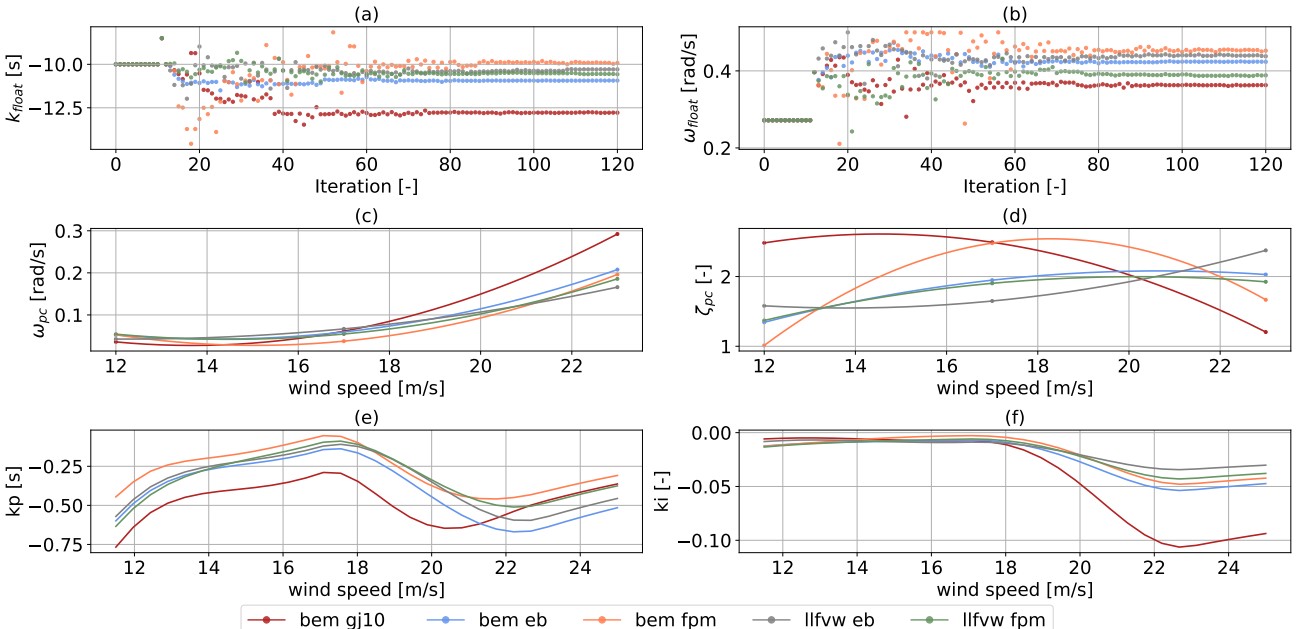

**Figure 8.** Convergence trends of design variables impacting the parallel compensation logic of the pitch controller to avoid negative damping (a, b), the pitch control tuning parameters (c, d) and the resulting proportional and integral gains (e, f).

Figure 8 displays the convergence trend of the feedback gain and low pass filter frequency of the parallel compensation logic
(a-b), the value of the three control points for the pitch controller bandwidth $\omega_{pc}$ and damping ratio $\zeta_{pc}$ (c-d) at the optimal iteration, as well as the optimized gain schedules for the proportional $k_{p,pc}$ and integral terms $k_{i,pc}$ of the collective pitch controller across the wind speed range (e-f). Generally, higher natural frequencies $\omega_{pc}$ reduce the rotor's response time, while larger damping ratios $\zeta_{pc}$ reduce the number of oscillations during the response (Abbas et al., 2022). The proportional gain schedule, which depends on both $\omega_{pc}$ and $\zeta_{pc}$, stands out for the BEM GJ10 case with larger absolute gains compared to the
other fidelity levels, which otherwise converge to a relatively close solution. The integral gain schedule, depending solely on $\omega_{pc}$, reveals a difference at high wind speeds, where the BEM GJ10 case shows a notable increase for above rated wind speeds, while the remaining fidelity levels maintain a flatter profile. As detailed by Abbas et al. (2024), high damping ratios allow for higher proportional gains and can help satisfy the overspeed constraint. Since all controllers were tuned with the same objective of minimizing tower base DEL, the resulting control parameters represent the optimal solution for a given platform, simulated
with the given aero-elastic fidelity level. Similar to the conclusion drawn by Zalkind et al. (2022) for platform-to-platform comparisons and the influence on a given platform type on the design of a tower, this approach enables a fair assessment of how modeling fidelity affects the platform–controller design process. Therefore, differences in the converged solution of the


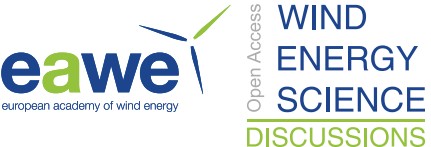

physical system (aka the floater), as presented in the following, can be attributed primarily to the influence of the aeroelastic fidelity on the optimization.

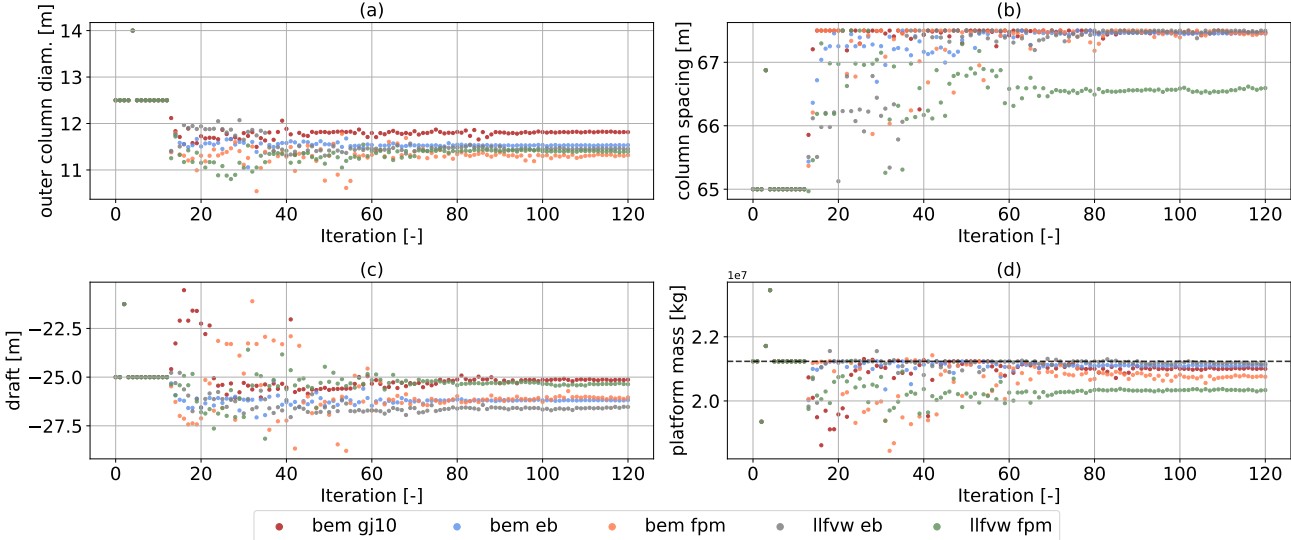

**Figure 9.** Convergence trends of design variables impacting the physical dimensions of the floating substructure and the resulting platform mass.

Figure 9 shows the evolution of the design variables related to the physical dimensions of the floating substructure (a-c) and the resulting platform mass (d). The platform mass of the initial iteration was set as a constraint. As a result, any increase in one geometric design variable must be offset by a decrease in others in order to comply within this constraint. Across all fidelity levels, the optimizer significantly reduced the outer column diameter (subplot a). This trend can be explained by hydrodynamic considerations, where decreasing the waterplane area reduces wave excitation and leads to lower dynamic loading. The BEM

GJ10 model converged to the largest final diameter, approximately 40 cm above the others. The remaining models are closely matched, with both FPM-based models converging to slightly smaller diameters than their Euler-Bernoulli counterparts. In contrast, column spacing (subplot b) was increased in all cases, reaching the upper bound for all models except for LLFVW FPM, which converged to a value approximately 1 meter below that limit. Increasing the column spacing increases the hydrostatic restoring moment. This compensates for the reduction of the restoring moment caused by reducing the outer column diameter.

No consistent trend is observed in the draft evolution (subplot c). It is worth stating that all configurations satisfied the platform mass constraint. The FPM-based models converged to lower mass levels than the EB-based ones. Interestingly, the LLFVW FPM configuration resulted in the lowest final platform mass, implying that a more efficient structural layout is enabled by the higher-fidelity aerodynamic representation.

Figure 10 presents a selected subset of the constraints that were set for the optimization problem. The AEP is plotted in
relative terms (subplot e), as each model starts from a different initial value. This respective initial AEP was imposed as a minimum constraint in each case to not allow the optimizer to trade a reduction in loads for a reduction in AEP. Subplot (f)





shows the blade root flapwise bending moment DEL, which was not explicitly included in the optimization problem, but is shown to illustrate that the found solutions did not lead to an increase in blade loading. The generator overspeed (subplot b) and the maximum platform pitch angle (subplot d) constraints are active across all fidelity levels and, in the case of BEM FPM, the overspeed constraint is not satisfied within 120 iterations, rendering the solution infeasible. The average pitch travel constraint (subplot c) is active only for the torsion-constraint case (BEM GJ10). Here, the absence of twist-to-feather load alleviation (aerodynamic and structural) in this case leads to increased pitch actuation, making this constraint more design-driving in comparison to the other levels of fidelity. The maximum platform pitch constraint is primarily driven by the reduction in outer column diameter. While increasing column spacing (as discussed in Figure 9) can offset the loss of hydrostatic restoring moment from smaller columns, the effect of aeroelastic fidelity becomes evident, since platform pitch is directly driven by rotor thrust and differences in predicted thrust between fidelity levels lead to different allowable column diameters.

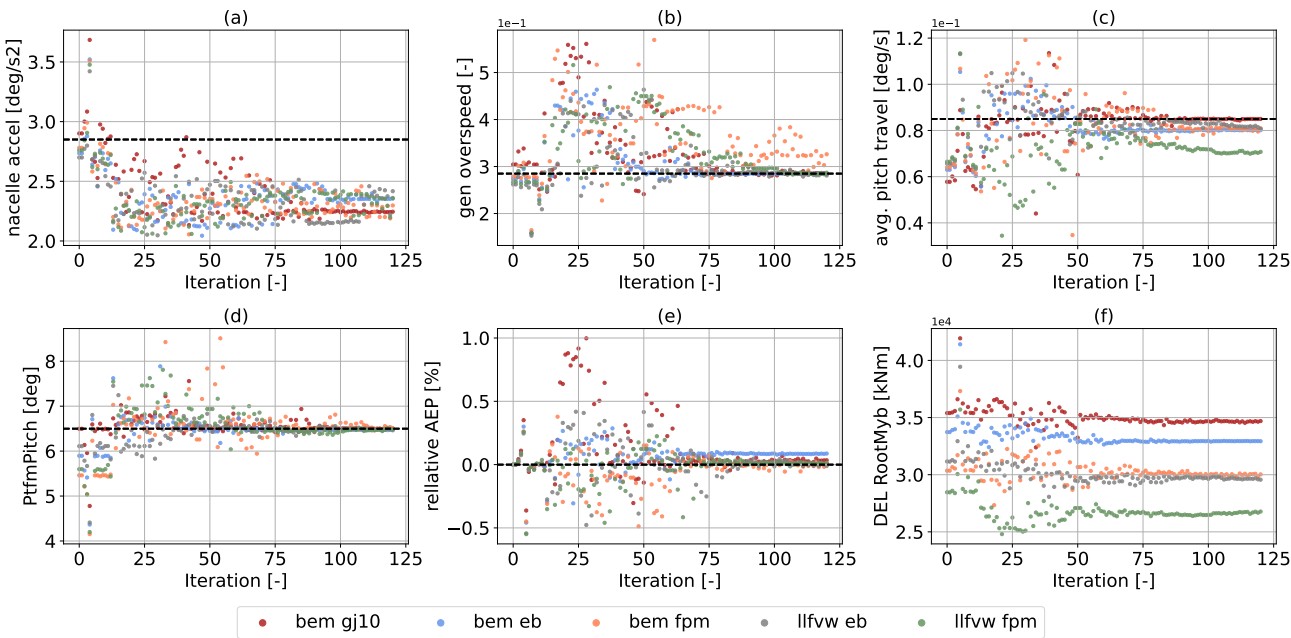

**Figure 10.** Convergence trends of the most relevant constraints, along with the flapwise DELs at the blade root. Although the blade root DELs were not part of the optimization problem, they confirm that none of the five cases resulted in increased load levels for the blades.

### 4.2.3 Frequency Domain Analysis

Figure 11 shows the power spectral densities (PSDs) for selected channels for the baseline (left column) and final iterations (right column) at an average wind speed of 13 m/s. The power spectrum of the tower base fore-aft bending moment (subplots a and b) is quite revealing concerning the energy at certain frequencies which were reduced by the optimizer. Focusing on the spectrum of tower base fore-aft moment for the baseline (subplot a), four distinct frequencies can be identified. The first one



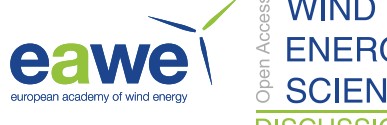

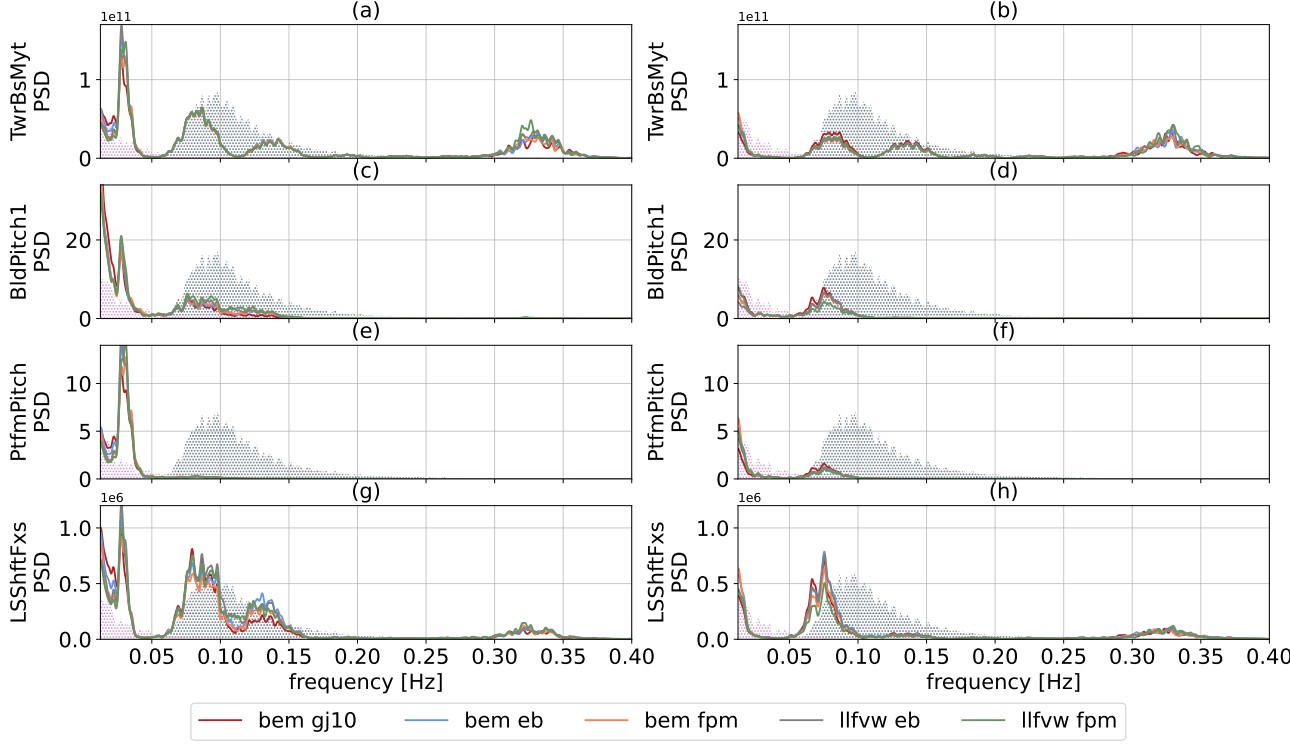

**Figure 11.** Power spectral densities of selected channels for the baseline and final iterations. The left column corresponds to the baseline and the right column to the final iteration for each method. The spectra are obtained from concatenated time series across all six seeds at a wind speed of 13 m/s. The dot-hatched areas represent the wind and wave spectra (scaled for visibility). The former is best seen between 0 and 0.05 Hz and the later between 0.05 and 0.2 Hz.

(from low to high frequencies) is present at the floater's natural pitch frequency ( 0.033 Hz). The next two frequency peaks lie within the linear wave frequency range and correspond to the excitation of the surge and pitch degrees of freedom by the linear waves. As these modes are phase-shifted and partially cancel each other out, resulting in the two-peak shape. The last one sits

around the 3P-frequency of the rotor. In subplot b, the corresponding spectra to the final iterations, it can be observed that the peak at the natural floater pitch frequency is completely eliminated. This is achieved mainly through the tuning of the floater bandwidth ($\omega_{pc}$), were a reduced bandwidth at corresponding wind speeds seems beneficial. The corresponding PSD's of the blade pitch actuation (c-d) reveal no actuation within the natural floater pitching frequency in the optimized solution, which is in contrast to the baseline. Further, the energy within the first peak within the linear wave frequency range (0.05–0.1 Hz), is

reduced by roughly half, while the second peak (0.11–0.16 Hz) remains largely unaffected. Given the multi-dimensional and nonlinear nature of the design space, it is difficult to draw definitive causal conclusions. However, the results indicate that this reduction is primarily achieved through a decrease in the waterplane area. Hence, one can observe that the solution with the largest outer column diameter (BEM GJ10) undergoes higher excitation in this frequency. A further reduction, albeit a smaller





gain compared to the reduction of waterplane area, within this frequency range can be explained by the blade pitch actuation (subplots c–d). Within the 0.05–0.1 Hz frequency range, slightly increased energy of blade pitch actuation is present in the final iteration - a result of the increased low-pass filter frequency (see Fig 8b). This leads to part of the wave excitation being dampened through the parallel compensation logic. This comes at a cost of increased platform pitching motion. Subfigures e-h are closely related to the blade pitch actuation, hence the peaks at platform natural frequency as well as the higher linear frequency range is reduced. Finally, the 3P excitation is also largely unaffected.

## 4.3 LCOE Optimization

From the analysis so far, there seem to be advantages related to increasing the level of aeroelastic fidelity in the design stage. In the previous optimization setup, however, where tower base loads were chosen as the merit figure and both AEP and platform mass were only treated as constraints, further reductions in mass or gains in AEP were not rewarded as long as the constraints were satisfied. As shown in Fig. 9, the LLFVW FPM case not only achieved a sizable reduction in loads but also lowered platform mass more than the other cases. For this reason, the optimization problem was reformulated with LCOE as the merit figure. In this setup, no constraints on mass, loads, or AEP were applied. Hence, the optimizer could search for the most cost-effective trade-offs between platform cost and AEP. The convergence trends of LCOE and its main contributors are shown in Fig. 12. All cases increased AEP and reduced platform cost. As already discussed, the same initial design (iteration 0), leads to variations in predicted AEP between different fidelity levels (Fig. 12c). The main trends are that LLFVW methods predict a larger AEP than BEM methods. As previously mentioned, this is a result of a chain of events that can be traced back to the induced velocities. LLFVW estimates smaller inductions than BEM. This increases the wind velocity in the rotor plane, leading the controller to increase the rotational speed in order to operate at an optimal tip speed. Ultimately, this results in an increase in mean thrust and, critically for AEP, torque (see Figs.5 and 6). Furthermore, as was discussed in Section 4.1.1, passive load alleviation (achieved through shear-twist coupling), which is only captured with the Timoshenko-FPM beams, affects AEP prediction and causes the observed differences in this metric. This carries through to the baseline LCOE values (Fig. 12a), which makes it difficult to draw a fair comparisons between the five cases. A regression analysis of sensitivities of LCOE with respect to AEP and platform cost reveals that for a 1% increase in AEP, the LCOE decreases almost 14 times more compared to a 1% decrease in cost. Consequently, even slight discrepancies in AEP prediction have the potential to significantly influence the LCOE outcome and mask reductions in platform mass (see Appendix A). Platform cost, in contrast to AEP, is not influenced by aeroelastic fidelity in the baseline design. Here, the three BEM-based optimizations demonstrate larger reductions in platform cost as the fidelity of the beam type increases (5.4% for GJ10, 6.0% for Euler–Bernoulli and 6.8% for FPM). The LLFVW-based cases achieved further reductions of 7% and 8.3% in platform cost respectively, also improving the result with increasing fidelity level of the beam type.

From Table 5, it is evident that all optimized cases reduced both the draft and the outer column diameter compared to the initial design. In contrast, column spacing varies between cases, which in turn drives the differences in platform mass. A key limiting factor was the heave period constraint (Fig. 13a), whose lower bound was active in all optimizations. This constraint was imposed to prevent the platform natural heave frequency to shift towards the linear wave frequency range.



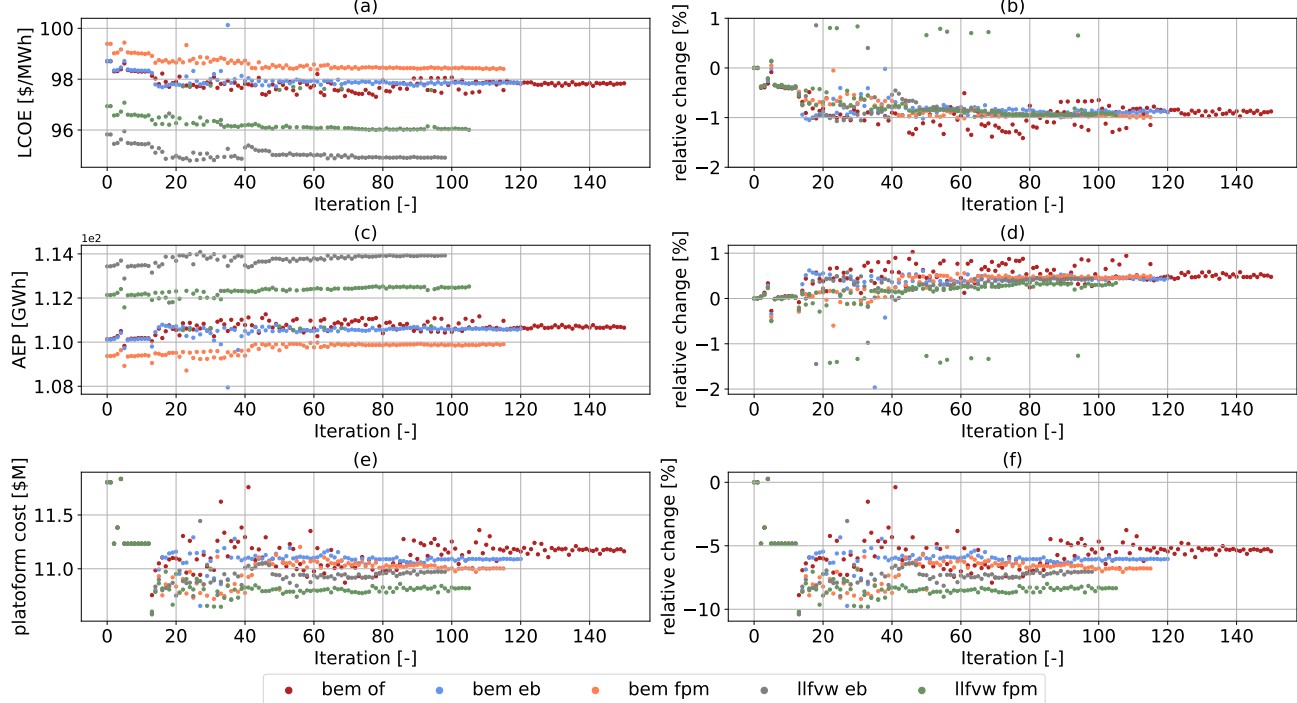

**Figure 12.** Convergence trends of absolute and relative levelized cost of energy (a) and (b), annual energy production (c) and (d) and platform cost (e) and (f). For the BEM-GJ10 configuration, the iteration limit was increased to 150, as convergence was not achieved within the initial limit of 120 iterations.

**Table 5.** Initial and final design variable values defining the dimensions of the floating substructure.

| Aerodynamic model | Beam model | draft [m] | outer column diam. [m] | column spacing [m] | platform mass [t] |
|---|---|---|---|---|---|
| initial | initial | 25.0 | 12.5 | 65.00 | 2.1235 e03 |
| BEM | GJ (10x) | 21.1 | 12.21 | 65.87 | 1.9160 e03 |
| BEM | Euler–Bernoulli | 20.0 | 12.11 | 67.49 | 1.8943 e03 |
| BEM | FPM | 20.0 | 12.0 | 66.92 | 1.8681 e03 |
| LLFVW | Euler–Bernoulli | 20.0 | 11.72 | 66.76 | 1.8593 e03 |
| LLFVW | FPM | 20.1 | 11.81 | 65.35 | 1.8119 e03 |

Table 6 lists the final design variables related to controller tuning. Across all cases, the floating feedback gain is reduced close to its lower bound (–8 s), while the low-pass filter frequency ($\omega_{float}$) shows more variation between models. The bandwidth

455    at wind speed control points 12 and 17 m/s, consistently is reduced to lower values, while it is elevated at 23 m/s. This mirrors the trends seen earlier in Fig. 8c, with the difference that this time BEM EB along with BEM GJ10 and LLFVW EB to a lesser extend, pushes the bandwidth higher compared to the remaining cases. The presumable cause for reducing bandwidth at 12 and



17 m/s is that the pitch controller reacts more slowly to above rated rotational speed and thereby favors AEP, while increasing bandwidth at 23 m/s is a requirement to comply with the generator overspeed, which is a constraint that is active across all cases (Fig. 13b).

**Table 6.** Initial and final value of design variables influencing the behavior of the servo controller.

| Aerodynamic model | Beam model | $k_{float}$ [s] | $\omega_{float}$ [rad/s] | $\omega_{pc}$ [rad/s] | $\zeta_{pc}$ [-] |
|---|---|---|---|---|---|
| initial | initial | 10 | 0.27 | [0.15\|0.15\|0.15] | [1.90\|1.90\|1.90] |
| BEM | GJ (10x) | −8.20 | 0.24 | [0.085\|0.043\|0.25] | [1.63\|2.50\|1.33] |
| BEM | Euler–Bernoulli | −8.11 | 0.16 | [0.065\|0.043\|0.26] | [1.40\|2.50\|2.49] |
| BEM | FPM | −8.00 | 0.32 | [0.049\|0.030\|0.12] | [1.63\|2.50\|2.35] |
| LLFVW | Euler–Bernoulli | −8.30 | 0.32 | [0.054\|0.047\|0.17] | [1.82\|2.50\|2.17] |
| LLFVW | FPM | −8.19 | 0.22 | [0.047\|0.036\|0.12] | [1.64\|2.37\|1.90] |

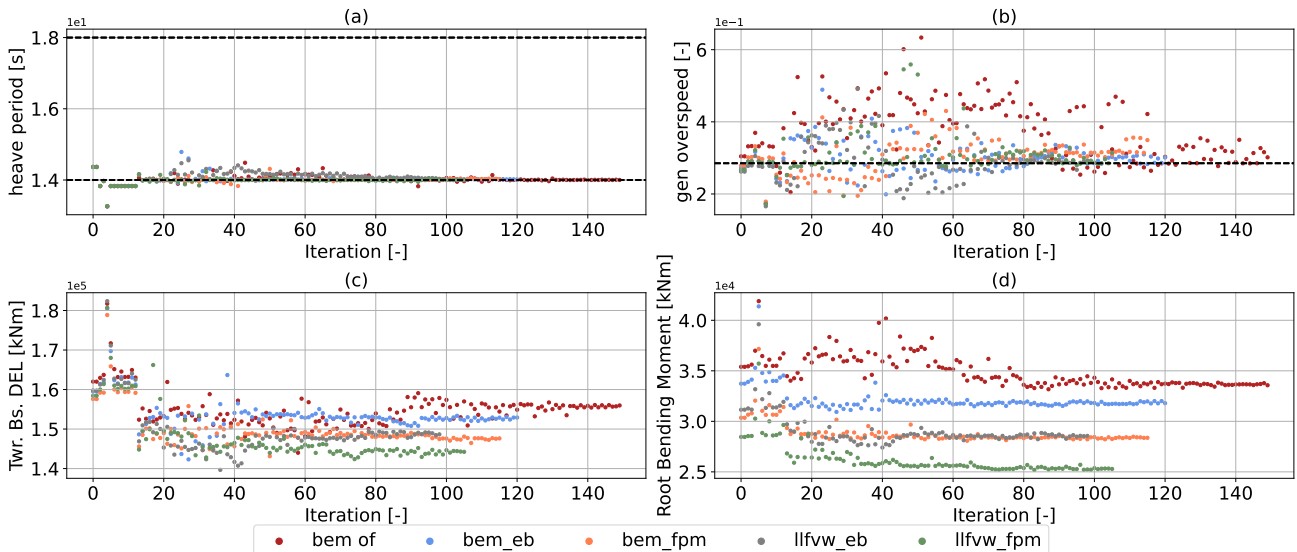

**Figure 13.** Active constraints during the optimization with merit figure LCOE along with damage equivalent loads of the tower base and blade root bending moment DELs.

The overspeed constraint appears to be particularly limiting in cases involving beam models that cannot capture the effect of structural shear-twist coupling (GJ10 and EB) and less so for the cases with Timoshenko-FPM beams. One possible reason for this is that, during unsteady events such as gusts, the increased blade loading causes a blade modeled with Timoshenko-FPM beams to twist toward feather due to shear–torsion coupling. This eases the load, which reduces the requirement on the pitch controller to respond rapidly and enables and operation with lower bandwidth, benefiting AEP. In contrast,GJ10 and EB models do not resolve this mechanism, resulting in higher required controller bandwidth. Interestingly, the bandwidth tuning





for the LLFVW EB case falls between the FPM and EB models at the control point of 23 m/s. This suggests that the choice of aerodynamic model also mitigates the overspeed constraint. Although it is somewhat surprising that the limited amount of wake-induced velocity influences controller tuning in these low tip speed ratio conditions, it is worth noting that the oscillation

occurs within a reduced frequency range of 1-2[4]. As laid out by Schulz et al. (2025), unsteady events in this reduced frequency range are primarily the dynamic wake effect and cause differences in rotor thrust amplitudes between BEM and LLFVW methods, with the latter predicting considerably smaller amplitudes. Even though tower base and blade root DELs were not part of the optimization problem, Figures 13c and d clearly show that increasing the modeling fidelity again yields beneficial results. As the level of aeroelastic fidelity increases, both tower and blade loads display stronger reductions.

## 475   5   Conclusion and Outlook

This work introduced QBtoWEIS, a new framework that integrates the QBlade and SONATA tools into the WEIS co-design optimization framework for floating offshore wind turbines. This integration adds two new levels of aeroelastic fidelity to WEIS, namely QBlade's unsteady polar blade element momentum and lifting-line free vortex wake methods. QBlade also provides a multi-body structural solver that uses one-dimensional beam representations, including both Euler-Bernoulli and

Timoshenko fully populated matrix elements. To enable the integrated generation of six-by-six stiffness and inertia matrices from the blade layup definition in WindIO, the cross-sectional analysis tool SONATA was also integrated into WEIS. This allows aerodynamic and structural modeling fidelity to be systematically varied within control co-design optimizations of floating offshore wind turbines. A comparative design study was carried out using QBtoWEIS to investigate the impact of aeroelastic fidelity on the controller and substructure designs of a modern wind turbine on a semi-submersible platform. Two

optimization problems were formulated and each was run with five different combinations of aerodynamic and structural fidelity levels. The first was a torsion-constrained Euler–Bernoulli beam model combined with an unsteady blade element momentum theory to mimic the level of fidelity already available in WEIS. Further, Euler–Bernoulli with torsion enabled and a Timoshenko-FPM beam model were paired with the blade-element-momentum or lifting-line free-vortex-wake code.

The first optimization problem was formulated to minimize the damage equivalent loads for the tower base moment by vary-

ing the dimensions of the floating substructure together with the pitch controller tuning parameters. The results confirmed that modeling fidelity choices can meaningfully influence key design-driving metrics, such as the rotor thrust and torque and subsequently platform pitch, blade pitch actuation and tower base fore-aft moment. Across the investigated fidelity combinations, the optimal designs produced different levels of tower base DEL reduction, as summarized in Table 7. A central mechanism that enabled reduced loads was the decrease in waterplane area, which was achieved by reducing the outer column diameter. The

maximum platform pitch constraint limited the minimum feasible diameter for each aeroelastic model. Hence, the influence of aeroelastic fidelity on rotor thrust directly affects the resulting platform design. Notably, the LLFVW FPM case converged to a design with considerably less platform mass than the BEM FPM case and the other configurations.

---

[4]The reduced frequency is a dimensionless number defined as $f_r = \frac{fD}{v_0}$, with f denoting the nacelle velocity frequency in fore-aft direction, D the diameter and $v_0$ the wind velocity parallel to the rotor axis (Schulz et al., 2025)





**Table 7.** Summary of the first optimization problem that aims to reduce tower base damage equivalent loads.

| Aerodynamic model | Beam model | rel. platform mass [%] | rel. DEL TwrBsMyt [%] |
|---|---|---|---|
| BEM | GJ (10x) | −1.1 | −10.71 |
| BEM | Euler–Bernoulli | −0.6 | −11.84 |
| BEM | FPM | −2.26 | −12.47 |
| LLFVW | Euler–Bernoulli | −0.35 | −13.08 |
| LLFVW | FPM | −4.23 | −12.76 |

The second optimization was a reformulation of the first problem with levelized cost of energy as the merit figure. Even though this metric proved not to be a perfect basis for comparison across fidelity levels, as differences in baseline AEP values differ significantly between models and AEP affects LCOE significantly more than platform cost, clear trends nevertheless emerged as shown in Table 8. In this regard, the LLFVW cases achieved the largest platform cost reductions (up to 8.3%), while the BEM-based cases showed smaller gains. Furthermore, the generator overspeed constraint appears to drive differences in controller tuning between the models. This constraint separates the BEM GJ10 and Euler-Bernoulli cases from the higher-fidelity models, suggesting that aeroelastic fidelity influences tuning.

**Table 8.** Summary of the second optimization problem that aims to reduce levelized cost of energy.

| Aerodynamic model | Beam model | rel. AEP [%] | rel. platform cost [%] | rel. LCOE [%] |
|---|---|---|---|---|
| BEM | GJ (10x) | 0.48 | −5.43 | −0.88 |
| BEM | Euler–Bernoulli | 0.44 | −6.04 | −0.89 |
| BEM | FPM | 0.49 | −6.79 | −0.99 |
| LLFVW | Euler–Bernoulli | 0.44 | −7.04 | −0.99 |
| LLFVW | FPM | 0.34 | −8.32 | −0.96 |

Higher modeling fidelity comes at a non-negligible computational cost. With the infrastructure used in this study, the LLFVW implementation in QBlade is approximately 12.5 times more expensive than the BEM method and runs around half the speed, while the Timoshenko-FPM beam model costs approximately 15% more than the Euler–Bernoulli formulation. The question whether an increase in modeling accuracy or a reduction in uncertainty justifies the additional expense has no straightforward answer. However, this work indicates that increasing the level of aeroelastic can lead to more efficient designs. This is mainly due to two mechanisms:

(i) Reduced conservativeness: Design-driving parameters such as damage equivalent loads or annual energy production differed considerably between the compared levels of fidelity, even for identical systems (iteration 0). Increasing the level of fidelity thus has the potential to reduce conservatism in the design.

(ii) Broader design space in constrained problems: the higher fidelity methods were able to navigate in a broader design space, relatively speaking, due to lower levels of thrust and torque (through shear-twist coupling) as well as reduced





standard deviations and min-to-max ranges of various quantities (caused by explicit modeling of the wake). This resulted in variations in both controller tuning and the sizing of the platform.

In contrast to other studies, where LLFVW methods could only be applied to a few, short and simplified cases, the efficient numerical implementation in QBlade enables its inclusion in the design phase. Further efficiency gains in numerical algorithms,

combined with the rapid global expansion of compute infrastructure should continue to reduce the relative expense of such models and potentially enable an increased usage of higher fidelity tools in future design processes. Furthermore, multi-fidelity approaches that combine the efficiency of lower-fidelity models with the accuracy of higher-fidelity methods, as demonstrated by Jasa et al. (2022), offer the possibility to include these methods in design processes. Future work will focus on exploring strategies to effectively combine LLFVW and BEM methods.



*Acknowledgements.* This work has received the support of the FLOATFARM Project, funded by the European Union's Horizon Europe research and innovation programme under grant agreement No. 101136091. Views and opinions expressed are however those of the author(s) only and do not necessarily reflect those of the European Union or the European Climate, Infrastructure and Environment Executive Agency (CINEA), which cannot be held responsible for them. The authors gratefully acknowledge the computing time made available to them on the high-performance computer "Lise" at the NHR center NHR@ZIB. This center is jointly supported by the Federal Ministry of Education and
Research and the state governments participating in the NHR (www.nhr-verein.de). During the preparation of this work, Artificial Intelligence (AI) through ChatGPT (GPT-5, OpenAI) and DeepL Write (DeepL SE) was employed solely for grammatical correction, stylistic refinement, and improvements in readability. At no stage were these tools used to generate, modify or verify any scientific results or methodological approaches. All substantive scientific ideas and proposed methodologies were developed independently by the authors without reliance on AI systems.

*Code and data availability.* For optimization, the QBtoWEIS interface was employed, which couples QBlade to the WEIS framework. The exact version of QBtoWEIS (v1.1.0) used in this study is archived at https://doi.org/10.5281/zenodo.14754213, while the maintained repository is available at https://github.com/rbehrensdeluna/QBtoWEIS.git. All simulations were carried out with the QBladeEE simulation environment (Enterprise Edition, for further information see https://qblade.org). A free Community Edition of QBlade, capable of running such optimizations, can be downloaded at https://qblade.org/downloads/.

*Author contributions.* The conceptualization, methodology, simulations, post-processing, data analysis and manuscript preparation were carried out by Robert Behrens de Luna. Francesco Papi supported the interpretation of results and contributed to manuscript editing. David Marten contributed through the development of the QBlade code, specifically enhancing compatibility with HPC execution and improving computational efficiency. Christian Oliver Paschereit provided supervision and feedback during the preparation of the work. All authors reviewed the manuscript.

*Competing interests.* The contact author has declared that neither he nor any of the co-authors have any competing interests.

**Nomenclature**

**AI** Artificial Intelligence

**AEP** Annual Energy Production

**BEM** Blade Element Momentum Theory

**CapEx** Capital Expenditure

**CCD** Control Co-Design

**COBYLA** Constrained Optimization by Linear Approximations

**CPU** Central Processing Unit

**EB** Euler–Bernoulli (beam theory)

**FEA** Finite Element Analysis

**FOWT** Floating Offshore Wind Turbine

**FPM** Fully Populated Matrix

**LCOE** Levelized Cost of Energy

**LLFVW** Lifting-Line Free Vortex Wake

**MDAO** Multidisciplinary Design Analysis and Optimization

**NREL** National Renewable Energy Laboratory

**NHR@ZIB** national high performance computing center at Zuse Institute Berlin

**PSD** Power Spectral Density

**RAO** Response Amplitude Operator

**RNA** Rotor–Nacelle Assembly

**SONATA** Structural Optimization and Aeroelastic Analysis

**WEIS** Wind Energy with Integrated Servo-control

**XDSM** (eXtended Design Structure Matrix





## Appendix A


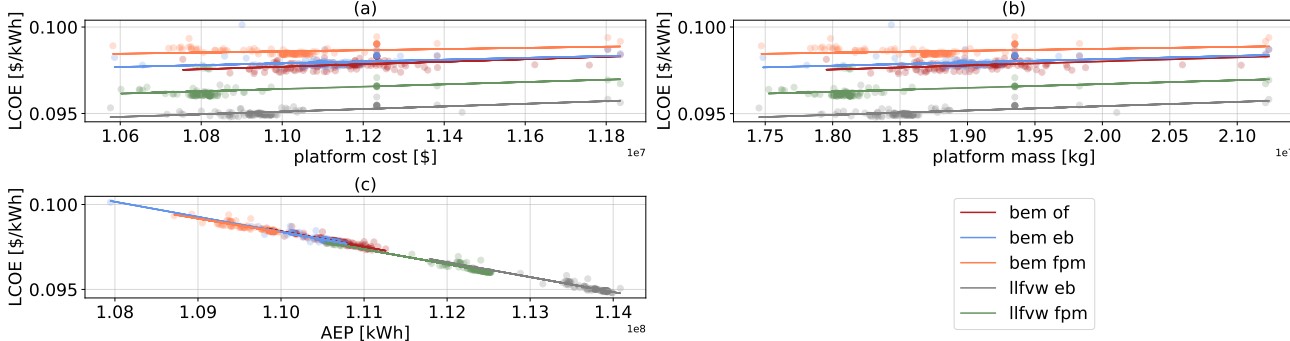

**Figure A1.** Linear regression fits showing the relationship between LCOE and platform cost (a), platform mass (b) and AEP (c).

As illustrated in Figure A1, the linear regression fits demonstrate the relationship between the levelized cost of energy and variations in platform cost, mass and AEP. The analysis indicates that even small changes in energy production have a significant impact on the LCOE metric. At the same time, the effect of platform cost and subsequently mass is very small. In order to quantify the relative importance of on the LCOE, the sensitivities regarding LCOE are calculated for the varying

measures. These sensitivites can be expressed through elasticity, which can be understood as the percentage change in an output (LCOE in this case) with respect to a percentage change in an input variable. For a generic input x and output y, the elasticity is can be defined as follows (Hamby, 1994):

$$b = \frac{\mathrm{d}Y}{\mathrm{d}X}\left(\frac{X}{Y}\right) \tag{A1}$$

To obtain the X and Y in Eq. A1, representative values for platform mass, cost and AEP (for the X) and LCOE (for the Y)

must be selected. For these representative values, the average was calculated across the varying aeroelastic fidelity levels from the initial iteration. The elasticities and further details concerning the linear regression are presented in Table A1.

**Table A1.** Linear-regression coefficients, fit statistics and elasticities of LCOE with respect to platform mass, platform cost and AEP, averaged across the five aeroelastic-fidelity optimizations.

|  | platform mass | platform cost | AEP | LCOE |
|---|---|---|---|---|
| Representative value | 21.2 [Mt] | 11.81 [M$] | 110 [GWh] | 97.9 [$/MWh] |
| Averaged $R^2$ | 0.12 | 0.12 | 0.88 | – |
| Averaged intercept | 93.4 [$/MWh] | 90.5 [$/MWh] | 194.4 [$/MWh] | – |
| Averaged slope | $2.014\,\mathrm{e}{-10}$ [$/(kWh·kg)] | $6.031\,\mathrm{e}{-10}$ [$/(kWh·$)] | $-8.736\,\mathrm{e}{-10}$ [$/(kWh$^2$)] | – |
| Averaged elasticity | 0.043 [%] | 0.073 [%] | -0.99 [%] | – |



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
