# Peer review of "Analyzing the Impact of Aeroelastic Model Fidelity on Control-Co Design Optimization of Floating Offshore Wind Turbines"

_Wind Energy Science, 2025_

## Referee Comment (RC1)

**Specific comments:**

**Abstract & Introduction**

- 1) "Higher fidelity models broaden the design space..." did you mean broaden the **feasible** design space?
- 2) "lead to more efficient platform designs" and "this can result in less efficient designs" efficient in which way(s)? Seems imprecise/vague, clarify if possible.
- 3) "enables the simultaneous variation of design variables across interacting subsystems and thus accounts for coupled physical phenomena" accounting for variation of design variables across subsystems is not sufficient to account for coupled physical phenomena disciplinary analysis submodels enable it.
- 4) "optimized and tuned" please clarify whether it is the platform being optimized and controlled tunned, or somehow both optimized and tuned.
- 5) "particularly for interactions between aeroelastic and control interactions" repetition
- 6) "especially in some of the dynamic conditions" vague
- 7) "comes with very little overhead" did you mean computational cost overhead?
- 8) "comparison of model fidelity levels included in QBlade within CCD optimization problems" what does it mean to compare fidelities within optimization? Might be worth rewording.

**Section 2.1**

- 1) "to ensure that resonant frequencies to not coincide with operational conditions." we can only talk about resonant frequencies about a pair of frequencies did you mean natural frequencies?
- 2) "enhances the design process" vague
- 3) Good high-level definition of CCD, however, could be useful to give an idea of how the controller and other subsystems design variables are being handled within the CCD problems, and example of which variables are typically included; this can help the readers appreciate the curse of dimensionality and why the traditional approach has been to handle the subsystems separately, or where the challenge lies.

**Sections 2.1-2.4**

- 1) "It has been validated and benchmarked" verified, if against other codes and not experiments.
- 2) "In contrast, WEIS includes" in contrast to what? Also, is the fact that WEIS has an option of a linear frequency-domain model relevant to this work? The next statement seems irrelevant too. This section could be streamlined for better readability.
- 3) "In order to obtain the equivalent beam parameters required for the Timoshenko-FPM beam model (i.e. the off-diagonal stiffness and inertia values),..." Good to explain what Timoshenko-FPM beam model is first.
- 4) "the fatigue loads at various design relevant channels of an onshore turbine" what are various design relevant channels?
- 5) "In (Papi et al., 2024), the authors confirmed similar findings..." would be good to elaborate on why the authors concluded that lifting line leads to better designs. What about the method/results led to it? Under/overprediction of which response led to what kind of conservatism in design?

- 6) "the wake method largely differs in the way of how the wake-induced velocities are calculated." did you mean "methods"?
- 7) "The bound vorticity of the blade is found by iteratively solving the Kutta-Joukowski theorem using estimates from 2D airfoil theory" could be explained more clearly
- 8) Equation 1: what is the star symbol? What is the lower case c?
- 9) "Problems like these are referred to as O(N2) problems." could be better to explicitly refer to problem complexity
- 10) Might be beneficial to give an idea of the computational cost in absolute terms: how long does it take to run a typical simulation? Also comparing between the different method introduced in this section, to better motivate the possible driver for using lowerfidelity models.
- 11) Modal reduction may need more explaining, as it seems to be one of the critical distinctions between the models considered here.
- 12) The last sentence of the section mentions the paper of Papi on impact of structural model fidelity on blade optimisation. Would be good to use it to position this work: how it expands on this previous research (because it seems to be a natural extension of it).
- 13) General comment: a figure or a table comparing the different models would be useful.

**Section 3**

- 1) "In practice, the exchange of information between these tools is limited to high-level data" vague.
- 2) When talking about the platform and controller usually designed separately, it might be worth to look into the literature. Since 2020 multiple papers have done multidisciplinary co-optimisation of the platform, tower and controller; A more thorough reference to MDAO for FOWTs research could be useful to set the background (rather than just referring to the review paper).
- 3) "all five optimizations" these have not been defined yet?
- 4) "To reduce the influence of transients, the initial conditions were set to an 11 m surge displacement and a 1° platform pitch angle." which environmental conditions does this refer to?
- 5) The 10 wind speed bind do not include the rated wind speed at which the turbine will spend most of its operational time. Does that impact the results/discussion in any significant way?

**Section 4**

- 1) Figure 7 is not introduced in the text.
- 2) Would it be better to split Figure 8 into two? It is confusing to have the two sets of subfigures with different x-axes.
- 3) "increasing aeroelastic fidelity improves not only the accuracy of load prediction but also enables more effective optimization" – this could only be stated if the initial and final designs resulting from each of 5 optimizations were evaluated with one consistent model – otherwise, it is impossible to compare which optimisation led to a better design.

- 4) The caption of Figure 8 should state that the 4 lower subplots are related to the optimised design.
- 5) "The platform mass of the initial iteration was set as a constraint" how was the initial iteration design chosen then? Do you refer to the starting point, or the design after 1 pass through the optimisation loop?
- 6) "Figure 10 presents a selected subset of the constraints that were set for the optimization problem" this wording is unclear, could imply that only a subset of constraints were considered in optimisation, please rephrase.
- 7) "The first combination, which constrains the torsional degree of freedom, represents the fidelity level provided by OpenFAST combined with ElastoDyn the current state of the art in WEIS. This analysis is followed by a discussion..." this reads as if only the first combination was to be presented, followed by the LCOE optimisation. Consider rewording.
- 8) The 3P region response seems to be relatively unaffected by the optimisation (in relation to Figure 11 and the corresponding discussion). Would including tower flexibility change anything in that respect?

**Section 4.3**

- 1) Would be good to include a table similar to Table 2b for clarity.
- 2) May be worth noting/discussing that the DEL minimisation did not clearly impact the draft, while LCOE minimisation did.

**Conclusion/general comments**

 How would the results compare to the case where blades elastically is ignored altogether? There is a lot of literature on FOWT optimisation with rigid body assumptions, and answering this question would be helpful in assessing these as well as recommending future directions.

---

## Author Comment (AC2)

**Response to Referee Comments - WES Discussions**

**Title:**

**Analyzing the Impact of Aeroelastic Model Fidelity on Control-Co Design Optimization of Floating Offshore Wind Turbines**

**Authors:**

Robert Behrens de Luna, Francesco Papi, David Marten, Christian Oliver Paschereit

Correspondence: Robert Behrens de Luna, Technische Universität Berlin (r.behrensdeluna@tu-berlin)

\*All Line and Figure references in the comments refer to the clean version (not LaTex Diff) of the revised manuscript.

**Reviewer 1:**

Very interesting and relevant research presented clearly and concisely. Although no new methods are proposed, state-of-the-art methods are used, with a new and valuable comparison of a few models and a demonstration of the impact of model choice on the optimisation process. Specific comments are included in the PDF attached. In general, the community will undoubtedly benefit from the new knowledge presented in this work.

Response: We would like to thank the reviewer for the positive and constructive feedback. The reviewer's suggestions were carefully considered and implemented wherever possible, which has led to an overall improvement in the quality and clarity of the manuscript. We appreciate the recognition of the relevance of this research.

**Specific Comments**

**Abstract & Introduction**

1. "Higher fidelity models broaden the design space..." – did you mean broaden the feasible design space?

**Response: Yes, thank you. This was updated in the abstract. [Line 11]**

2. "lead to more efficient platform designs" and "this can result in less efficient designs" – efficient in which way(s)? Seems imprecise/vague, clarify if possible.

Response: The term "less efficient" referred to the optimization's objective. We agree that the original phrasing was not clear enough and have revised it. The sentence was removed from the abstract since the following statement already addresses this point. In the introduction, we updated the wording for clarity. [Line 19 ff.]

3. "enables the simultaneous variation of design variables across interacting subsystems and thus accounts for coupled physical phenomena" – accounting for variation of design

variables across subsystems is not sufficient to account for coupled physical phenomena – disciplinary analysis submodels enable it.

Response: The sentence was rephrased to clarify the point made by the reviewer. [Line 22 ff.]

4. "optimized and tuned" – please clarify whether it is the platform being optimized and controlled tunned, or somehow both optimized and tuned.

Response: The wording has been updated to clarify that both the physical system and the controller are optimized simultaneously. The term "tuned" was removed for clarity as controller tuning parameters are treated as design variables within the optimization. Further, similar occurrences in the manuscript were updated accordingly as well. [Line 27]

5. "particularly for interactions between aeroelastic and control interactions" – repetition

Response: Sentence was updated in the manuscript [Line 29]

6. "especially in some of the dynamic conditions" – vague

Response: Sentence was updated in the manuscript, now stating the conditions investigated in the quoted sources. [Line 37]

7. "comes with very little overhead" – did you mean computational cost overhead?

Response: The sentence was clarified, now stating that the overhead indeed refers to computational cost. [Line 56 ff.]

8. "comparison of model fidelity levels included in QBlade within CCD optimization problems" – what does it mean to compare fidelities within optimization? Might be worth rewording.

Response: We agree that the original phrasing was misleading. The sentence has been revised to make it clear that the comparison refers to CCD optimization results obtained using different levels of aeroelastic model fidelity implemented in QBlade. [Lines 60-61]

**Section 2.1**

1. "to ensure that resonant frequencies to not coincide with operational conditions." – we can only talk about resonant frequencies about a pair of frequencies – did you mean natural frequencies?

Response: The term "resonant frequencies" was replaced with "natural frequencies" and the sentence was slightly rephrased and corrected. [Lines 79-80]

2. "enhances the design process" – vague

Response: The wording was removed and the sentence slightly rephrased for clarity. [Lines 84]

3. Good high-level definition of CCD, however, could be useful to give an idea of how the controller and other subsystems design variables are being handled within the CCD problems, and example of which variables are typically included; this can help the readers appreciate the curse of dimensionality and why the traditional approach has been to handle the subsystems separately, or where the challenge lies.

Response: We thank the reviewer for this suggestion. A description of the types of design variables typically included in CCD problems was added to this section, along with a reference to a recent tower—controller CCD study. [Lines 86-89]

**Section 2.1-2.4**

1. "It has been validated and benchmarked" – verified, if against other codes and not experiments.

Response: The sentence was revised to clarify between validation with experiments and benchmarking against other simulation tools. [Lines 99-101]

2. "In contrast, WEIS includes" – in contrast to what? Also, is the fact that WEIS has an option of a linear frequency-domain model relevant to this work? The next statement seems irrelevant too. This section could be streamlined for better readability.

Response: The sentence was revised so that the reference to RAFT is removed to streamline the section.

3. "In order to obtain the equivalent beam parameters required for the Timoshenko-FPM beam model (i.e. the off-diagonal stiffness and inertia values),..." — Good to explain what Timoshenko-FPM beam model is first.

Response: We thank the reviewer for the comment. A short clarification was added to the section that the description of the modeling approaches is provided in following sections. [Lines 105-106]

The subsection "QBlade in the WEIS Framework" focuses on the coupling between tools and the integration of QBlade and SONATA within WEIS and including detailed model descriptions here would, in the opinion of the authors, shift the focus away from that objective.

4. "the fatigue loads at various design relevant channels of an onshore turbine" – what are various design relevant channels?

Response: The sentence was modified to clarify that sensors representative of the turbine's load response are meant, and the specific channels were explicitly named. [Lines 126-129]

5. "In (Papi et al., 2024), the authors confirmed similar findings..." – would be good to elaborate on why the authors concluded that lifting line leads to better designs. What about the method/results led to it? Under/overprediction of which response led to what kind of conservatism in design?

Response: We thank the reviewer for this comment. However, as our objective in this section is to summarise existing findings which motivate this current work rather than reproduce

their detailed analysis, we have chosen to provide a short description and direct interested readers to the referenced work for a comprehensive explanation of the underlying mechanisms. To better guide the reader, the manuscript now mentions the load sensors considered (e.g. blade-root and tower-base bending moments) [Line 128] and clarifies that the overprediction of these loads by the BEM method (as shown by the referenced work) motivates the use of the higher-fidelity LLFVW method. Further discussion of differences in the modeling approaches are covered later in the Results (Section 4, Lines 313 ff.)

6. "the wake method largely differs in the way of how the wake-induced velocities are calculated." – did you mean "methods"?

Response: The sentence was modified to clarify that, in the context of this work, the influence of the different wake methods on the resulting loads stems from the way each method calculates the wake-induced velocities. [Lines 144-145]

7. "The bound vorticity of the blade is found by iteratively solving the Kutta-Joukowski theorem using estimates from 2D airfoil theory" – could be explained more clearly

Response: The paragraph was rephrased to more clearly explain how the process to determine the blade's bound circulation works. Readers interested in learning more about the LLFVW method can refer to the provided sources. [Lines 156 ff.]

8. Equation 1: what is the star symbol? What is the lower case c?

Response: The star symbol was a typographical error and should be interpreted as a standard multiplication sign. The variable c, which stands for the chord length, was added to the equation description. [Eq. 1]

9. "Problems like these are referred to as O(N2) problems." – could be better to explicitly refer to problem complexity

Response: We thank the reviewer for the suggestion. The sentence was rephrased to explicitly state that  $O(N^2)$  refers to the computational complexity of evaluating the Biot–Savart law for all vortex elements. [Lines 172 - 173]

10. Might be beneficial to give an idea of the computational cost in absolute terms: how long does it take to run a typical simulation? Also comparing between the different methods introduced in this section, to better motivate the possible driver for using lowerfidelity models.

Response: General comparisons of computational cost are highly dependent on factors such as model discretization, wake settings, and computational infrastructure. We therefore believe this point is appropriately addressed in section 3.3 "Computational Considerations and Infrastructure" and Table 4, where the reader can see the computational cost for this particular presented optimization case and has already been introduced to the underlying physical models and the number of simulations required per iteration.

11. Modal reduction may need more explaining, as it seems to be one of the critical distinctions between the models considered here.

Response: We thank the reviewer for this comment. We have expanded the brief overview of the modal representation used in ElastoDyn, and refer interested readers to the cited references for further details. [Lines 192 ff.]

12. The last sentence of the section mentions the paper of Papi on impact of structural model fidelity on blade optimisation. Would be good to use it to position this work: how it expands on this previous research (because it seems to be a natural extension of it).

Response: We thank the reviewer for this suggestion. A sentence was added to clarify how this work expands upon prior research on structural model fidelity in wind turbine blade optimization [Lines 205 ff.]. Specifically, it highlights that the present study extends these ideas to the system level by examining how variations in beam and wake model fidelity influence platform sizing and controller tuning within a control co-design framework [Line 209 ff.]. As the Papi et al. 2025 reference is still under review, it it was removed from the manuscript

13. General comment: a figure or a table comparing the different models would be useful.

Response: The authors agree that a concise table that summarizes the models improves clarity. A summary table has been added to Section 2.4 (Table 1), which provides an overview of the aerodynamic and structural models implemented in QBlade, their main assumptions and captured effects.

**Section 3**

1. "In practice, the exchange of information between these tools is limited to high-level data" – vague.

Response: We agree with the reviewer that the original phrasing was vague. We revised the sentence to specify that the information typically exchanged between tools includes global load envelopes and thrust coefficients, rather than detailed aeroelastic data .[Lines 224 ff.]

2. When talking about the platform and controller usually designed separately, it might be worth to look into the literature. Since 2020 multiple papers have done multidisciplinary co-optimisation of the platform, tower and controller; A more thorough reference to MDAO for FOWTs research could be useful to set the background (rather than just referring to the review paper).

Response: We appreciate the reviewer's suggestion to expand the discussion of recent MDAO studies. However, we believe that this context is already sufficiently covered in the Introduction, where several key references on CCD and MDAO for FOWTs are discussed and quoted.

- → Zalkind and Bortolotti (2024) Control Co-Design Studies for a 22 MW Semisubmersible Floating Wind Turbine Platform
- → Garcia-Sanz (2019) Control Co-Design: An Engineering Game Changer
- → Zalkind et al. (2022) Floating Wind Turbine Control Optimization
- → Yu et al. (2024) Control Co-Design Optimization of Floating Offshore Wind Turbines with Tuned Liquid Multi-Column Dampers

- → Bayat et al. (2025) Nested Control Co-Design of a Spar Buoy Horizontal-Axis Floating Offshore Wind Turbine
- → Abbas et al. (2024) Control Co-Design of a Floating Offshore Wind Turbine
- → Ojo and Collu (2022) Multidisciplinary design analysis and optimization of floating offshore wind turbine substructures: A review.

The reference to Zalkind and Bortolotti (2024) in this section was not intended to reintroduce the broader research background. Instead, we want to indicate that the optimization problem in our work adapts the problem formulation used in that study. Therefore, we maintained the current level of background detail in this section to focus on describing the optimization setup.

3. "all five optimizations" – these have not been defined yet?

Response: We thank the reviewer for pointing this out. The sentence was revised [Line 257] to clarify that all optimizations used identical QBlade models, except for the selected wake and beam models. The detailed introduction of the five optimization cases is provided in Section 4 (Results, Table 4).

4. "To reduce the influence of transients, the initial conditions were set to an 11 m surge displacement and a 1° platform pitch angle." – which environmental conditions does this refer to?

Response: The sentence was revised to clarify that the specified initial conditions were applied consistently across all wind speed bins and selected as suitable initialization states for all simulations. [Line 264 ff.]

5. The 10 wind speed bin do not include the rated wind speed at which the turbine will spend most of its operational time. Does that impact the results/discussion in any significant way?

Response: We thank the reviewer for the observation. According to "Zahle — Definition of the IEA Wind 22-Megawatt Offshore Reference Wind Turbine" and as stated in Table 2 of the manuscript, the rated wind speed of the IEA 22 MW reference turbine is 11 m/s. This speed is included in the set of 10 wind speed bins. Thus, the rated operating condition is already included in the analysis. In our opinion, even if neighboring wind speed bins had been selected instead, the corresponding operating condition would still be sufficiently represented, as turbulent wind fields frequently cover this region across the number of seeds (see the figure below, which shows the wind velocity for the six seeds simulated at the 11 m/s wind speed bin).

**Section 4**

1. Figure 7 is not introduced in the text.

Response: Figure 7 is now introduced in the text with a short sentence describing what it shows. [Line 363]

2. Would it be better to split Figure 8 into two? It is confusing to have the two sets of subfigures with different x-axes.

Response: We thank the reviewer for the helpful comment. Figure 8 was split up into two figures (now 8 and 9). The describing text of the figure has been modified accordingly.

3. "rediction but also enables more effective optimization" – this could only be stated if the initial and final designs resulting from each of 5 optimizations were evaluated with one consistent model – otherwise, it is impossible to compare which optimisation led to a better design.

Response: We thank the reviewer for pointing this out, and we have added a subsubsection on a cross-comparison, including the figure below (Section 4.2.4 "Cross-Evaluation of Design Outcomes"). This comparison shows how the final outcomes of the different optimizations compare when analyzed with the same aeroelastic model.

4. The caption of Figure 8 should state that the 4 lower subplots are related to the optimised design.

Response: Thank you for the suggestion. This is now clearly stated in the caption of the figure (Now) Figure 9.

5. "The platform mass of the initial iteration was set as a constraint" — how was the initial iteration design chosen then? Do you refer to the starting point, or the design after 1 pass through the optimisation loop?

Response: The starting point is meant. The sentence was revised and now states "initial design" instead of "initial iteration". [Line 390]

6. "Figure 10 presents a selected subset of the constraints that were set for the optimization problem" — this wording is unclear, could imply that only a subset of constraints were considered in optimisation, please rephrase.

Response: The sentence was rephrased to clarify that the figure (now Fig. 11) shows only a selection of the constraints, while the full set is provided in Table 3. [Line 403]

7. "The first combination, which constrains the torsional degree of freedom, represents the fidelity level provided by OpenFAST combined with ElastoDyn — the current state of the art in WEIS. This analysis is followed by a discussion..." — this reads as if only the first combination was to be presented, followed by the LCOE optimisation. Consider rewording.

Response: The sentence was rephrased to clarify that five cases were run, and the first combination represents the current state-of-the-art fidelity level with OpenFAST and ElastoDyn. The remaining cases incrementally increase the aeroelastic fidelity. [Lines 291 ff]

8. The 3P region response seems to be relatively unaffected by the optimisation (in relation to Figure 11 and the corresponding discussion). Would including tower flexibility change anything in that respect?

Response: In fact, tower flexibility is included in all of the cases that are presented. A clarification was added to specify that the tower is being modeled as a flexible structure (Section 3.2 Modeling Considerations). [Lines 263-264]

**Section 4.3**

1. Would be good to include a table similar to Table 2b for clarity.

Response: We thank the reviewer for the suggestion. Since the reformulated optimization differs only by the removal of two constraints and the modification of the merit figure, we opted not to add a mostly redundant table. Instead, the text now explicitly refers to Table 2b and specifies which two constraints were omitted in the LCOE optimization. [Line 460]

2. May be worth noting/discussing that the DEL minimisation did not clearly impact the draft, while LCOE minimisation did.

Response: We agree with the point made by the reviewer and have added a discussion before Table 5, explaining that LCOE optimization drives draft reduction through economic benefits of lower material costs, while DEL optimization found increased draft beneficial for load mitigation. [Lines 481 ff.]

**Conclusion/general comments**

1. How would the results compare to the case where blades elastically is ignored altogether? There is a lot of literature on FOWT optimisation with rigid body assumptions, and answering this question would be helpful in assessing these as well as recommending future directions.

Response: We thank the reviewer for raising this point. In our view, including a rigid-body case in the present study would not be suitable to provide future directions on the raised matter, given the used set-up. Our framework relies on a fully coupled, nonlinear, time-domain aeroelastic model, in which structural flexibility is fundamental to the

simulation model. The only way to mimic a rigid turbine in this environment would be to considerably increase the stiffness in all degrees of freedom. However, the outcome of such an optimization would not represent the class of reduced-order or frequency-domain optimization methods mentioned by the reviewer and would therefore offer limited additional value, in our opinion.

Nevertheless, the topic of how a rigid-body assumption compares to flexible or fully coupled formulations has been partially addressed by Zalkind and Bortolotti (2024). They demonstrated the benefits of fully coupled MDAO—CCD optimization over sequential processes, in which the floating substructure is dimensioned using a linear frequency-domain solver, and then the aero-, servo-, hydro-, and elastic OpenFAST model is leveraged to control the time domain in a second step.

Our work builds upon these findings by focusing specifically on the influence of aeroelastic model fidelity within a fully coupled, nonlinear framework and is not suited for comparing flexible and rigid representations.

**Reviewer 2:**

This article presents a design optimization study using QBlade with multiple levels of modeling fidelity. The authors clearly demonstrate the influence of model fidelity on optimization outcomes and design results. While the findings are well presented and valuable to the community, the practical implications for design decision-making remain somewhat unclear. Addressing a few underlying assumptions and design philosophy questions would strengthen the paper.

Response: We thank the reviewer for the thoughtful and constructive feedback. We appreciate the recognition of the study's value as well as the valuable suggestions to improve its practical implications and underlying design assumptions. These comments were helpful in strengthening the manuscript. We have tried to address each point carefully in the following responses.

**Major Comments**

**Major Comment 1**

**Reviewer:**

The manuscript could more clearly guide readers on how to apply these findings in practice. For instance, should designers favor higher-fidelity models because they yield less conservative results, or rely on lower-cost models while accounting for uncertainty through safety factors? Given the current reliability challenges of offshore turbines, should the design community aim to reduce conservatism and safety margins? Alternatively, should model fidelity be driven by the need to accurately capture specific design features, like the ability to represent blade torsion in bend–twist coupled blades?

Response: We thank the reviewer for this comment. We agree that the relationship between model fidelity, conservatism, and design safety margins is an important topic.

In our view, the primary objective when addressing reliability challenges for floating offshore wind turbines is to minimize modeling uncertainty. Once the designer has high confidence that the simulation results accurately represent the physical behavior, safety factors can be applied to those results to account for uncertainty in materials, manufacturing and design. Hence, in our opinion, safety margins are not determined by the level of model fidelity itself, but rather by the predicted response of a design that is as close to reality as possible. In the present study, increasing aeroelastic fidelity did not necessarily reduce conservatism in a structural sense. Rather, it modifies the design space in a way that an optimizer e.g., finds different, and in this case, better performing, optimal substructure dimensions compared to when using lower-fidelity methods. This was true even when the designs were reevaluated in a cross-comparison (see our response to major comment 4).

In the end, it is the opinion of the authors that the decision on how to apply safety factors is independent of the chosen model. In fact, higher-fidelity models that predict smaller loads based on a more accurate physical description enable, in the presented case-study, a smaller, more cost-efficient substructure without compromising reliability. The counterfactual would also hold true, if higher-fidelity models were to predict increased loads, the resulting designs would be less cost-efficient but presumably more robust against real-world conditions. It is

our opinion that the overall objective is to improve the accuracy and confidence in the models we rely on during the design process.

We acknowledge that the final statement in the abstract and some statements in the conclusion were potentially misleading. They have been revised accordingly. We want to emphasize that the findings of this work should not be interpreted to recommend reducing safety margins.

**Major Comment 2**

**Reviewer:**

Section 2.3, particularly its introduction, appears central to interpreting later results. A summary table or schematic comparing the effects and assumptions of the different model fidelities could improve clarity and accessibility.

**Response:**

The authors agree that a concise table that summarizes the models improves clarity. This point was also mentioned by the other reviewer. A summary table has been added in form of Table 1, which provides an overview of the aerodynamic and structural models implemented in QBlade, their main assumptions and captured effects.

**Major Comment 3**

**Reviewer:**

The study uses only DLC 1.1 for design optimization. Could the authors comment on why other design load cases (e.g., DLC 1.6 or 6.1) were excluded, given that prior work has shown their significant influence on common design constraints?

**Response:**

The metocean conditions used in this study correspond to a site located near the Isle of Barra in Scotland and are particularly severe. In analysis carried out in preparation for this work, the extreme sea states defined in DLC 1.6 have shown to be too extreme and prevented normal turbine operation. Performing a CCD under these conditions would not have led to representative results. The purpose of the present work is not to obtain a fully optimized floating substructure design but rather to isolate and assess the influence of aeroelastic model fidelity on the optimization outcome in representative conditions. Zalkind and Bortolotti (2024) showed that controller tuning parameters were mostly consistent between optimizations running DLC 1.1 and DLC 1.6, the most noticeable difference being the floating feedback gain. They hence recommend DLC 1.6 for similar studies. Given the severity of the current site, we therefore believe that focusing on DLC 1.1 is consistent with the intent of their recommendation.

Furthermore, because our merit figure is based on a fatigue-related load metric, the use of a DLC representative of normal power production seemed reasonable. DLC 6.1 was not included, since the rotor operates in an idling condition, and thus control parameter tuning has no effect on the design process, making it irrelevant for the primary intention of the paper. We acknowledge, however, that expanding the DLC set would be a valuable extension for future work, especially when design suggestions sit at the core of the work.

**Major Comment 4**

**Reviewer:**

It might be informative to compare the results of a low-fidelity optimized design when re-evaluated using a higher-fidelity model.

Response: We thank the reviewer for pointing this out. This point was also mentioned by the other reviewer. We have added a cross-comparison in section 4.2.4 [Line 441] that includes the figure below. This comparison shows how the final outcomes of the different optimizations compare when analyzed with the same aeroelastic model.

**Major Comment 5:**

**Reviewer:**

Since models are ultimately approximations of reality, do the authors assume that higher-fidelity models are more accurate representations of real designs? If so, what evidence or validation supports that assumption?

Response: We thank the reviewer for this comment. We agree that all numerical models are approximations of reality. However, the assumption that higher-fidelity models are more accurate representations of reality is based on their more complete physics based representation.

In the context of aerodynamics, the lifting-line free vortex wake methods explicitly resolve wake dynamics. In contrast, the blade element momentum method relies on empirical corrections. Studies comparing the two methods have shown that discrepancies can be traced back to simplified treatment of certain physical phenomena (Perez-Becker et al., 2020; Papi et al., 2024, Schultz et al., 2025) that become ever more important with increasing rotor sizes. Furthermore, the literature also includes experimental validation that demonstrates that vortex-based methods agree more closely with measured data, particularly under unsteady conditions (Boorsma et al., 2016; Bergua et al., 2023).In the context of structural dynamics, using more complete beam models that resolve coupled dynamics between degrees of freedom is a requirement to resolve effects such as the often referred to bend-twist or shear-twist coupling. This is also acknowledged in the technical report on the design of the IEA22MW wind turbine.

Based on this, we believe it is a valid and well-supported assumption that higher-fidelity models provide a more physically accurate representation of aeroelastic behavior.

**Minor Comments**

Reviewer: The tower mass in Table 1 appears inconsistent—perhaps a comma/decimal point issue. Please verify the value.

Response: We thank the reviewer for pointing this out. The decimal point was removed to align the value in (now) Table 2 with the IEA 22MW wind turbine documentation provided by "Zahle et al. - Definition of the IEA Wind 22-Megawatt Offshore Reference Wind Turbine".

Reviewer: Consider merging Tables 3 and 4 for improved readability.

Response: We thank the reviewer for the suggestion. Tables 3 and 4 from the original manuscript have been combined into one table (now) Table 4, which summarizes the aeroelastic fidelity levels and their associated computational costs. We updated the surrounding text to reflect these changes.

Reviewer: The manuscript states that the Frobenius norm is used to compute DELs, which differs from standard fatigue damage calculations and is typically applied to matrices. Please clarify or correct this methodology.

Response: We agree that the original phrasing was misleading. The Frobenius norm is not applied to compute the DELs directly but rather to combine the three time series of the tower-base bending moments (fore–aft, side–side, and torsion) into a single resulting load time series for the tower base, before the fatigue analysis. The DELs are then computed in the standard way via rainflow counting on this combined channel using pCrunch. This is the standard procedure to calculate this load channel in the WEIS framework as well. The footnote was modified to clarify this distinction and avoid confusion. [Page 13]

Reviewer: Around line 412, the term "floater bandwidth" is described as  $\omega_pc$ , though earlier  $\omega_pc$  refers to the pitch control bandwidth. Are you instead referring to a coupled platform—pitch frequency? Please elaborate.

Response: We thank the reviewer for finding this error. Indeed the pitch control bandwidth was meant in this context. The manuscript has been updated accordingly. [Line 427]

Reviewer: In Figure 11, it would be helpful to mark or annotate the dominant platform, tower, and rotor natural frequencies.

Response: We agree with the reviewer that this makes the figure and related analysis easier to interpret. The figure (now Figure 12) has been updated accordingly to include annotations for the dominant platform, tower, and rotor natural frequencies.

Reviewer: In Table 6, the signs of the initial and final values for k\_float appear to differ. Please confirm whether this is intentional or a typographical error.

Response: We thank the reviewer for noticing this. The sign inconsistency in the values of k\_float in Table 6 was a typographical error and has been corrected in the revised manuscript. [Page 25]